# Srs2 promotes synthesis-dependent strand annealing by disrupting DNA polymerase δ-extending D-loops

Jie Liu[1], Christopher Ede[1†], William D Wright[1], Steven K Gore[1], Shirin S Jenkins[1], Bret D Freudenthal[2‡], M Todd Washington[3], Xavier Veaute[4], Wolf-Dietrich Heyer[1,5*]

[1]Department of Microbiology and Molecular Genetics, University of California, Davis, Davis, United States; [2]Department of Biochemistry, University of Iowa Carver College of Medicine, Iowa City, United States; [3]Department of Biochemistry, University of Iowa Carver College of Medicine, Iowa City, United States; [4]DRF-IRCM-CIGEx, CEA, Fontenay aux Roses, France; [5]Department of Molecular and Cellular Biology, University of California, Davis, Davis, United States

*For correspondence: wdHeyer@ucdavis.edu

Present address: †Department of Chemical and Biomolecular Engineering, University of California, Los Angeles, Los Angeles, United States; ‡Department of Biochemistry and Molecular Biology, University of Kansas Medical Center, Kansas City, United States

Competing interests: The authors declare that no competing interests exist.

**Abstract** Synthesis-dependent strand annealing (SDSA) is the preferred mode of homologous recombination in somatic cells leading to an obligatory non-crossover outcome, thus avoiding the potential for chromosomal rearrangements and loss of heterozygosity. Genetic analysis identified the Srs2 helicase as a prime candidate to promote SDSA. Here, we demonstrate that Srs2 disrupts D-loops in an ATP-dependent fashion and with a distinct polarity. Specifically, we partly reconstitute the SDSA pathway using Rad51, Rad54, RPA, RFC, DNA Polymerase δ with different forms of PCNA. Consistent with genetic data showing the requirement for SUMO and PCNA binding for the SDSA role of Srs2, Srs2 displays a slight but significant preference to disrupt extending D-loops over unextended D-loops when SUMOylated PCNA is present, compared to unmodified PCNA or monoubiquitinated PCNA. Our data establish a biochemical mechanism for the role of Srs2 in crossover suppression by promoting SDSA through disruption of extended D-loops.

## Introduction

Loss of heterozygosity (LOH) can be an important contributor to carcinogenesis and a principal mechanism leading to LOH is crossover formation by homologous recombination (HR) (*Knudson, 2001*; *Stern, 1936*). HR is a high-fidelity DNA repair pathway for DNA double-strand breaks (DSB) and interstrand cross-links (*Heyer et al., 2010*). HR is also involved in the recovery of stalled or collapsed replication forks, as well as in the bypass of damage in the DNA template. In general, there are two types of HR repair products, crossovers and non-crossovers, depending on whether the arms of sister chromatids or homologs are exchanged or not, respectively. Crossovers between non-allelic DNA regions lead to chromosomal aberrations including deletions, inversions, and trans-locations. Non-crossover is the preferred outcome of HR in somatic cells avoiding potential loss of heterozygosity and possible chromosome rearrangements. In contrast, during meiosis HR is designed to generate at least one crossover per homolog to facilitate accurate chromosome segregation during the first meiotic division (*Hunter, 2015*). The D-loop is the joint molecule produced by the Rad51-DNA filament after homology search and DNA strand invasion, key steps of the HR process (*Heyer et al., 2010*). DNA polymerases, in particular DNA polymerase δ, extend the invading 3'-OH in the D-loop (*McVey et al., 2016*). The extended D-loop is a critical intermediate in

determining crossover/non-crossover outcome. In the double Holliday junction (dHJ) pathway, where both DSB ends are engaged to form a joint molecule with two Holliday junctions (HJs), both crossover and noncrossover products can be formed by nucleolytic resolution, while enzymatic dissolution leads to non-crossover products only. In break-induced replication, recombination-associated DNA synthesis extends for the rest of the chromosome arm directly causing LOH. During Synthesis-Dependent Strand Annealing (SDSA), the newly extended D-loop is disrupted to produce exclusively non-crossover products, thus avoiding the possibility for LOH. SDSA is the preferred HR pathway in somatic cells and is likely dynamically regulated to ensure faithful DNA repair while avoiding LOH and chromosome rearrangements.

Multiple enzymes can act on D-loops either to abort recombination (anti-recombination) or influence the repair outcome (crossover/non-crossover) (*Heyer et al., 2010*). The first DNA strand invasion product is the nascent D-loop and reversion of this HR intermediate represents a mechanism of anti-recombination. Several proteins have been shown to disrupt protein-free, nascent D-loops in vitro, including Sgs1 and its human homolog BLM (*Bachrati et al., 2006*; *Fasching et al., 2015*; *van Brabant et al., 2000*). Rad54 is required for D-loop formation in yeast and stimulates D-loop formation by the human RAD51 protein (*Ceballos and Heyer, 2011*). Both yeast and human Rad54 can also dismantle D-loops made with short invading ssDNAs in reconstituted in vitro reactions containing Rad51 and RPA. This is especially true for invading ssDNA lacking flanking dsDNA regions such as found in a resected DSB (*Bugreev et al., 2007a*; *Wright and Heyer, 2014*). Moreover, Top3-Rmi1, a type1A topoisomerase that associates with Sgs1 (human BLM-TOPOIIIα-RMI1/2 is the homologous complex) efficiently dissolves Rad51-mediated nascent D-loops in reconstituted reactions, an activity that is shared by the human TOPOIIIα-RMI1/2 complex (*Fasching et al., 2015*) and appears to be physiologically relevant (*Kaur et al., 2015*; *Tang et al., 2015*). Finally, human RTEL-1 can reverse RAD51-mediated D-loops and may act in the specific context of telomeres in dismantling T-loops (*Vannier et al., 2012*).

Although quite similar in DNA structure, the extended D-loop represents a very different target, and dismantling this HR intermediate is pro-recombinogenic affecting the crossover/non-crossover outcome. Three major helicases have been genetically implicated in this process in the budding yeast *Saccharomyces cerevisiae*: Sgs1, Mph1, and Srs2. Current evidence suggests that all three act in distinct but possibly partially overlapping pathways (*Ira et al., 2003*; *Mitchel et al., 2013*; *Prakash et al., 2009*). These helicases represent the prototypes for the homologous or analogous activities in all eukaryotes including the human proteins BLM (yeast Sgs1), FANCM (yeast Mph1), and FANCJ, FBH1, PARI, RECQ1, RECQ5, RTEL-1 (proteins with analogous activities to Srs2) (*Heyer et al., 2010*; *Sung and Klein, 2006*). Sgs1 (human BLM) is involved at multiple stages in HR as part of a complex with Top3-Rmi1 (human TOPOIIIα-RMI1/2) including DSB end-resection and dHJ dissolution (*Bizard and Hickson, 2014*; *Cejka et al., 2010*; *Niu et al., 2010*). Mutations in Sgs1 increase the crossover/non-crossover ratio, and this effect is explained by the role of Sgs1 in dHJ dissolution, as a defect in dHJ dissolution provides opportunity for nucleolytic resolution of dHJs and crossover formation (*Ira et al., 2003*). Mph1 (human FANCM) is a 3′−5′ DNA helicase required for recombination-dependent error-free bypass of DNA damage during DNA replication (*Prakash et al., 2005*; *Schürer et al., 2004*). Genetic evidence shows that Mph1 contributes to the SDSA pathway, and the Mph1 protein is able to disrupt nascent and extended D-loops during in vitro reconstituted reactions (*Prakash et al., 2009*; *Sebesta et al., 2011*).

The *SRS2* gene was originally discovered as a suppressor of a *rad6* mutation (*SRS2*=*S*uppressor of *R*AD *S*ix) and equally suppresses a defect in the *RAD18* gene, which encode the Rad6-Rad18 ubiquitin E3 ligase that targets PCNA for mono-ubiquitination to favor tranlesion DNA synthesis (*Hoege et al., 2002*; *Lawrence and Christensen, 1979*). Srs2 is a member of the SF1 helicase family and translocates on ssDNA with a 3′ to 5′ polarity using the energy from ATP hydrolysis (*Rong and Klein, 1993*). The Srs2 ATPase activity is required for both its activities in HR (anti-recombination, pro-SDSA; see below). The Srs2 helicase represents the prototypic anti-recombinase dismantling the Rad51-ssDNA filament, which performs the central HR reactions of homology search and DNA strand invasion (*Krejci et al., 2003*; *Veaute et al., 2003*). In addition, accumulating genetic evidence also reveals a positive role of Srs2 in HR, specifically in the formation of non-crossover products by the SDSA pathway (*Aylon et al., 2003*; *Dupaigne et al., 2008*; *Ira et al., 2003*; *Mitchel et al., 2013*; *Miura et al., 2013*; *Robert et al., 2006*; *Saponaro et al., 2010*). In an HO-induced ectopic recombination assay, *srs2Δ* cells show a significant decrease in the level of non-crossover products;

however, the level of crossovers remains almost unchanged, leading to an increase of percentages of crossovers among the total products (*Ira et al., 2003*). Similarly, in a spontaneous ectopic recombination assay, *srs2Δ* cells show a four-fold increase in crossover formation such that almost half of the recombination products are crossovers (*Robert et al., 2006*). Post-translational modifications of PCNA are involved, as suggested by the genetic analysis of PCNA mutants defective in ubiquitylation/sumoylation, of the SUMO ligase Siz1, and the ubiquitin-conjugating enzyme Rad6, all of which shows increase in crossover levels in a plasmid-based SDSA assay (*Miura et al., 2013*). The involvement of SUMO-PCNA in crossover suppression is also confirmed in other ectopic recombination systems (*Burkovics et al., 2013*; *Robert et al., 2006*). In sum, these data provide compelling evidence for an additional, pro-recombinogenic role of Srs2. It was proposed that Srs2 disrupts extended D-loops to favor annealing with the second end during SDSA to generate non-crossover products (*Ira et al., 2003*; *Robert et al., 2006*).

While the mechanism of anti-recombination by Srs2 is well established, the mechanisms involved in its pro-recombination (SDSA) role and its function in adaptation/recovery remain to be determined (*Macris and Sung, 2005*; *Marini and Krejci, 2010*; *Vaze et al., 2002*). Biochemical characterization of Srs2 demonstrated that it unwinds various synthetic replication and recombination intermediates including replication forks, 3'- and 5'- flaps, Holliday junction (HJ), and nicked HJ, but not D-loops in reconstituted recombination reactions (*Le Breton et al., 2008*; *Marini and Krejci, 2012*; *Prakash et al., 2009*; *Sebesta et al., 2011*). It was proposed that Srs2 competes with Pol δ for PCNA binding to inhibit HR-associated DNA synthesis, and by this SDSA, in an ATP-independent mechanism (*Burkovics et al., 2013*). However, later genetic work showed that the Srs2 ATPase activity is essential for its SDSA activity (*Kolesar et al., 2016*; *Miura et al., 2013*), suggesting that competition for PCNA binding could be an aspect but not the underlying fundamental mechanism. The apparent lack of D-loop disruption activity of Srs2 also suggested a second possible mechanism for its SDSA activity by dissociating Rad51 from the second end of the DSB to enable annealing with extended invading strand from the disrupted D-loop (*Marini and Krejci, 2010*; *Mitchel et al., 2013*).

Here, we report that Srs2 is capable in vitro to disrupt Rad51/Rad54-produced D-loop structures mimicking physiological length and structures, providing biochemical evidence for a specific SDSA mechanism by Srs2. The SDSA pathway was partially reconstituted with RPA, Rad51, Rad54, RFC, DNA Polymerase δ and three different but fully modified forms of PCNA. Consistent with the genetic data, we demonstrate that Srs2 disrupts D-loops in an ATP-dependent fashion with a slight but significant preference for extending D-loops over unextended D-loops when SUMO-PCNA is present, compared to unmodified PCNA or monoubiquitinated PCNA (Ubi-PCNA). Our data provide a plausible mechanism of Srs2 activity in SDSA and assign a specific role of PCNA sumoylation during HR.

## Results

### Srs2 disrupts D-loops mimicking physiological length in a time- and concentration-dependent manner

*S. cerevisiae* Rad51 relies on Rad54 to generate D-loop structures. Typically in reconstituted systems, short ssDNAs (~35–100 nts) are used for Rad51 filament formation and DNA strand invasion, however the D loops products in such reactions suffer from instability due to Rad54's ability to dismantle these short D loops efficiently (*Bugreev et al., 2007b*; *Wright and Heyer, 2014*). This complicates data analysis and was found to be unsuitable to explore the D-loop disruption activity of Srs2 (*Burkovics et al., 2013*; *Prakash et al., 2009*). Thus, there is a need to generate stable D-loop structures in reconstituted reactions. To this end, we utilized a series of long ssDNA substrates, which have been designed to resemble physiological-length Rad51 filaments that produce relatively stable D-loop structure (*Wright and Heyer, 2014*). An ssDNA with 607 nucleotides (named '607' thereafter) of homology to a supercoiled dsDNA donor is able to form D-loops with high yield (~50%) and stability over the time course of a D-loop reaction (*Figure 1A,B*). In addition, we carefully staged the reaction by adding Srs2 10 min after the initiation of D-loop formation by Rad54 and dsDNA substrate, which allows accumulation of maximal D-loop levels (*Wright and Heyer, 2014*). This protocol avoids the complication of overlaying the phase of D-loop formation with D-loop dissociation

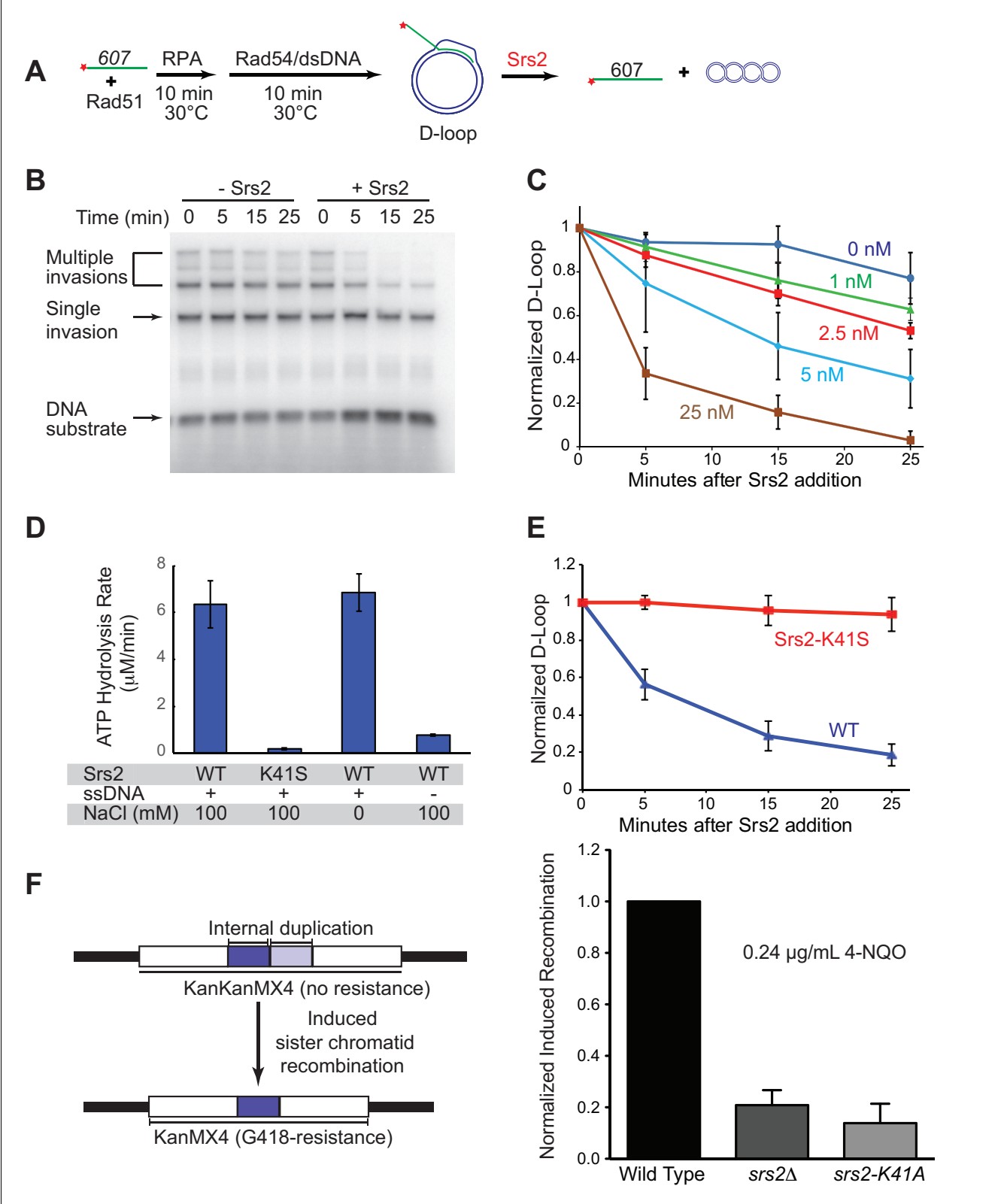

**Figure 1.** Srs2 disrupts D-loops in a concentration-, time-, and helicase activity-dependent manner. (**A**) D-loop disruption assay. Fully homologous ssDNA with 607 nucleotides was used as substrates in B and C. Rad51 (0.2 μM) was incubated with 1 nM ssDNA substrate (Rad51/nt = 1:3) for 10 min at 30°C. RPA (33 nM) was added for another 10 min incubation before the addition of 84 nM Rad54 and 7 nM supercoiled plasmid dsDNA (21 μM bp). (**B**) Time course of Srs2 titration into pre-formed D-loops produced by Rad51 and Rad54. Srs2 (0 or 5 nM) was added and incubated for various time at

*Figure 1 continued on next page*

*Figure 1 continued*

30°C. Samples were taken at 0, 5, 15, and 25 min and quenched by SDS/Proteinase K treatment. (**C**) Quantitation of normalized total D-loop yield in (**B**) and additional titrations. Absolute initial D-loop yields were between 50% and 75%. (**D**) Srs2-K41S is deficient in ATP hydrolysis. Rates of ATP hydrolysis were determined for 5 nM wild type Srs2 (WT) or catalytic-deficient Srs2-K41S (K41) in D-loop reaction buffer containing either 0 or 100 mM NaCl, with or without 10 μM φX174 ssDNA as cofactor. (**E**) Time course of D-loop disruption by 5 nM of either wild type Srs2 (WT) or catalytic-deficient Srs2-K41S (K41S). Plotted are means ± standard deviation from n = 3. Absolute initial D-loop yields were between 44% and 72%. (**F**) Induced sister chromatid recombination assay. The normalized (wild type = 100%) induced recombination frequencies are shown at 0.24 μg/mL 4-NQO as means ± standard deviations from three experiments for wild type, *srs2△*, and *srs2-K41A* (see *Table 1*). The frequency of spontaneous $G418^R$ cells was subtracted from the induced recombination frequencies. The full 4-NQO dose response is shown in *Figure 1—figure supplement 2*.

The following source data and figure supplements are available for figure 1:

**Source data 1.** Source data for *Figure 1C*.
**Source data 2.** Source data for *Figure 1D*.
**Source data 1.** Source data for *Figure 1E*.
**Source data 4.** Source data for *Figure 1F* and *Figure 1—figure supplement 2A–D*.
**Figure supplement 1.** D-loop disruption by Srs2 does not depend on RPA.
**Figure supplement 1—source data 1.** Source data for *Figure 1—figure supplement 1*.
**Figure supplement 2.** ATPase activity of Srs2 is required for both pro- and anti-recombination function and required for 4-NQO-induced recombinational repair, and responsible for the extreme UV and MMS sensitivities of *rad18△* strain.

catalyzed by Srs2. Previous attempts to identify D-loop disruption activity by Srs2 used reaction schemes where Srs2 was added immediately after or simultaneously with Rad54 and dsDNA substrate (*Burkovics et al., 2013*; *Prakash et al., 2009*). The combination of staged reactions and long ssDNA substrates enabled us to identify robust dissociation of D-loops by Srs2 in a time- and concentration-dependent manner (*Figure 1B,C*). As previously observed (*Wright and Heyer, 2014*), *607* can invade multiple dsDNA molecules to produce D-loops, where one ssDNA invaded up to three duplex DNA molecules (*Figure 1B*). All D-loop species from single invasion to multiple invasions were disassembled almost completely by 25 nM Srs2 in a 25 min reaction (*Figure 1C*). At intermediate concentrations (5 nM in *Figure 1B*) it appears as if multiple invasion products are more sensitive to Srs2. However, multiple invasions are processed into single invasions before complete disruption, making it impossible to assess whether multiple invasions or single invasions are preferentially disrupted. As little as 2.5 nM Srs2 showed significant D-loop disruption in these reactions that contained 1 nM invading ssDNA, showing that Srs2 is highly effective in D-loop disruption. D-loop disruption by Srs2 is not dependent on the presence of the single-stranded DNA binding protein RPA (*Figure 1—figure supplement 1*). We conclude that Srs2 is capable of disrupting nascent D-loops, and below we show directly that Srs2 is also capable of disrupting D-loops that are being extended by DNA polymerase δ, the presumed in vivo substrate in the SDSA pathway.

## The translocase/helicase activity of Srs2 is required for D-loop disruption

In plasmid-based and chromosomal assays, the ATPase activity of Srs2 is required for normal levels of non-crossover formation (*Kolesar et al., 2016*; *Miura et al., 2013*). To determine whether the ability to translocate or unwind dsDNA is required for Srs2 to disassemble D-loops, we purified an ATPase-defective Srs2 point mutant carrying a single amino acid substitution in the Walker-A motif (Srs2-K41S). Srs2-K41S displays no detectable ATP hydrolysis (*Figure 1D*), while wild type Srs2 shows strong, ssDNA-stimulated ATP hydrolysis. As expected, the motor/helicase activity of Srs2 was required to dissociate D-loops (*Figure 1E*), since the addition of Srs2-K41S did not alter D-loop level. The Srs2-K41S mutant protein appears to fold properly as it purifies like the wild type protein (not shown) and interacts with Rad54 like wild type Srs2 (Figure 3E), suggesting that it is the lack of

ATPase and not incorrect protein folding that leads to the D-loop disruption defect. Moreover, the Srs2 motor activity is required for 4-Nitroquinine-N-oxide (4-NQO)-induced recombination (*Figure 1F*), an assay that reports on Rad51-dependent sister-chromatid recombination (*Ede et al., 2011*). As expected, deletion of *RAD51* is epistatic to deletion of *SRS2* for induced recombination, and induced recombination is strongly Rad51-dependent (*Figure 1—figure supplement 2A–C*). Additionally, mutation of *SRS2* increases the spontaneous recombination rate in this assay system and this increase is abolished by a *rad51* mutation (*Figure 1—figure supplement 2D*). This result is consistent with previous observations that show a hyper-rec effect for *SRS2* deletion mutations in spontaneous recombination (*Palladino and Klein, 1992*). Finally, the Srs2 motor activity is also required for anti-recombination in vivo as indicated by the suppression of the severe sensitivity of *rad18△* strain towards UV and methyl methanesulfonate (MMS) by the *srs2-K41R* and *srs2-K41A* mutations (*Figure 1—figure supplement 2E*). *rad18△* suppression by these *SRS2* Walker A box mutations was not evident in a different assay of UV-sensitivity (*Burkovics et al., 2013*). Together these results show that the translocase/ATPase activity of Srs2 is required for its two essential functions in HR, Rad51 filament disruption (anti-recombination) and D-loop disruption (pro-SDSA/anti-crossover). These results are consistent with previous findings that the Srs2 motor activity is essential for its anti-recombination activity (*Krejci et al., 2004*) and pro-SDSA function (*Kolesar et al., 2016*; *Miura et al., 2013*).

## Srs2 is capable of disrupting different D-loop structures

During long-range DSB resection, Sgs1/Dna2 or Exo1, degrade the 5′ strand to produce 3′ tailed ssDNA spanning hundreds of nucleotides, which represents the in vivo substrate for Rad51 filament formation (*Mimitou and Symington, 2008*; *Zhu et al., 2008*). After showing that Srs2 efficiently disrupts D-loops, we wanted to test further whether Srs2 shows any substrate preference towards D-loops produced from ssDNA with zero (*607*), one (98ds-*607*, 3′-tailed), or two (98ds-*607*-78ds) terminal duplex heterologies mimicking different cellular repair scenarios of DSB repair (3′-tailed) or gap repair (*Figure 2A*). These D-loops with terminal heterologies add another layer of stability against Rad54 disruption in addition to increased ssDNA length that enables longer heteroduplex (hDNA) to form (*Wright and Heyer, 2014*). The yield of D-loop was slightly different with all the three substrates, but Srs2 dismantled all D-loops efficiently (*Figure 2B,C*). After normalizing the D-loop level according to the initial yield, it became clear that Srs2 displays a significant preference toward D-loops made with 3′-tailed invading DNA (98ds-*607*) (*Figure 2D*). This preference is confirmed by further experiments determining the preferred orientation of D-loop disruption (see below, Figure 4). We conclude that Srs2 is capable of disrupting different nascent D-loop structures with modest preference for D-loops containing 3′-tailed invading DNA, the mimic for physiological DSB repair.

## Srs2 disrupts Rad51/Rad54 produced D-loops more effectively than protein-free D-loops

Next, we wanted to discern the specificity of Srs2 towards D-loops as a DNA intermediate and identify any effects of the proteins required for D-loop formation on D-loop disruption. To this end, we generated protein-free D-loops and compared the disruption by Srs2 with D-loops formed in reconstituted reactions with the cognate Rad51 and Rad54 proteins. First, we used Rad51/Rad54 to generate *607*-based D-loop structure at high yield conditions and purified protein-free D-loops by removing all proteins. Then, we tested whether these protein-free D-loops can serve as substrates for Srs2. The reactions contained about 85% of protein-free D-loops, and the results are plotted as normalized D-loops with the initial level set at 100% to allow comparison. As shown in *Figure 3B*, 15 nM Srs2 disrupts only about 35% of the available protein-free D-loops in 40 min, while Srs2 at similar enzyme/DNA ratios disrupts almost all D-loops at 25 min in the Rad51/Rad54 reconstituted reaction (*Figures 1C*, 25 nM >95%, 5 nM ~70%). Using this data (*Figure 1C*, *Figure 3B*), we estimate an about ten-fold difference in the disruption between protein-free D-loops and D-loops in the reconstituted reaction. Thus, although Srs2 recognizes and loads onto D-loop as a protein-free DNA structure, the sheer presence of Rad51 and Rad54 enables more effective disruption by Srs2. Srs2 interacts with Rad51 physically through its C-terminal interaction site (*Colavito et al., 2009*), and this protein-protein interaction has been demonstrated to trigger ATP turnover and dissociation of

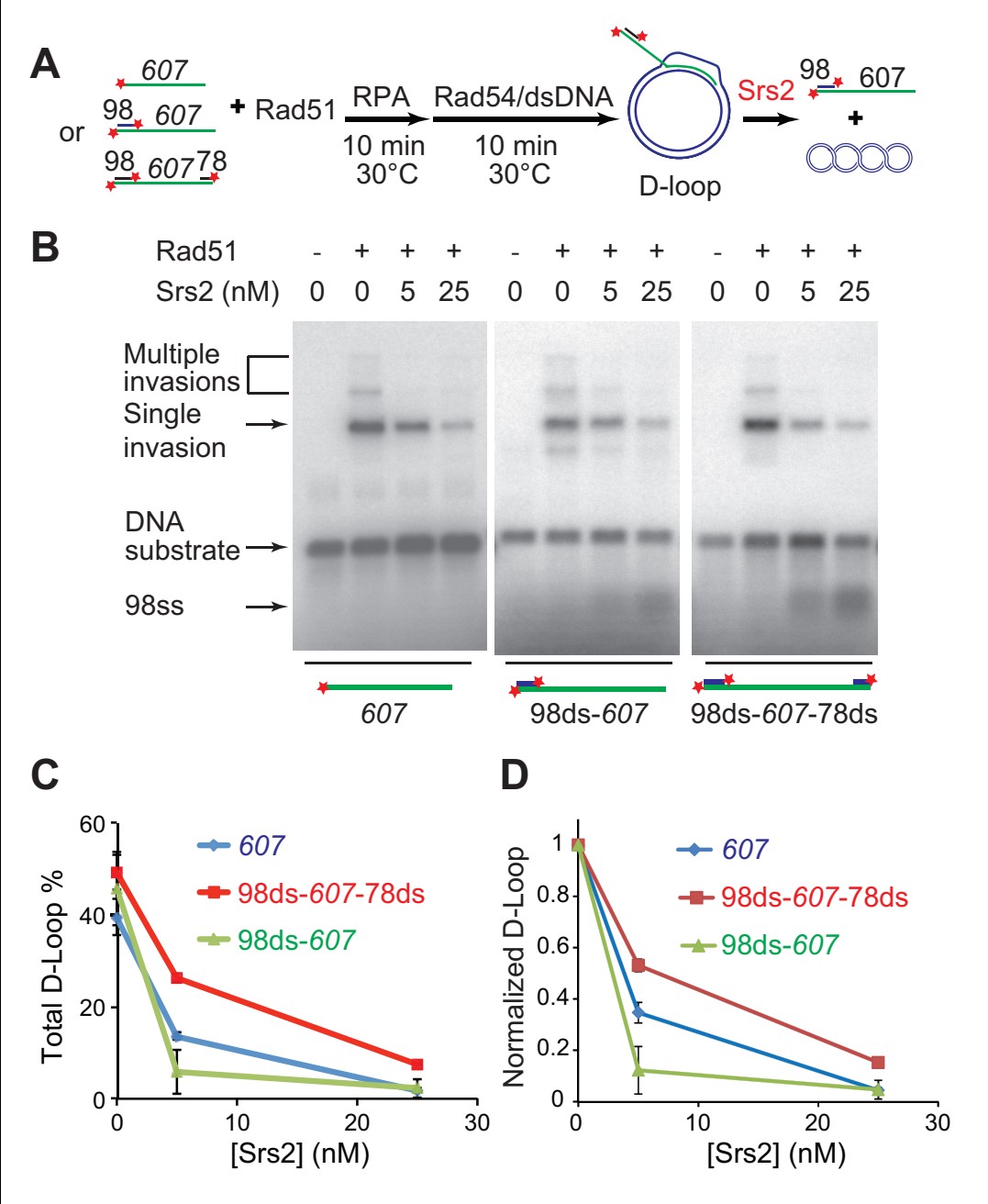

**Figure 2.** Srs2 disrupts various D-loops produced from ssDNA substrates mimicking physiological length and structure. (**A**) D-loop disruption assay and substrates. Rad51 (0.2 μM) was incubated with 1 nM ssDNA substrates (Rad51/nt = 1:3) for 10 min at 30℃. 33 nM RPA was added for another 10 min incubation before the addition of 84 nM Rad54 (84 nM) and 7 nM supercoiled plasmid dsDNA (21 μM bp). Srs2 (0, 5, or 25 nM) was added and incubated for 10 min at 30℃ before the reaction was stopped by SDS/Proteinase K. (**B**) Srs2 titration in D-loop disruption assays with three different DNA constructs, 3' tailed DNA with 5' heterology, ssDNA with full homology, gapped DNA with both 5' and 3' heterologies. (**C**) Quantitation of total D-loops from (**B**). (**D**) Quantitation of normalized D-loops from (**B**) setting initial D-loop yield as 100%. Plotted are means ± standard deviation from n = 3.

The following source data is available for figure 2:

**Source data 1.** Source data for *Figure 2*.

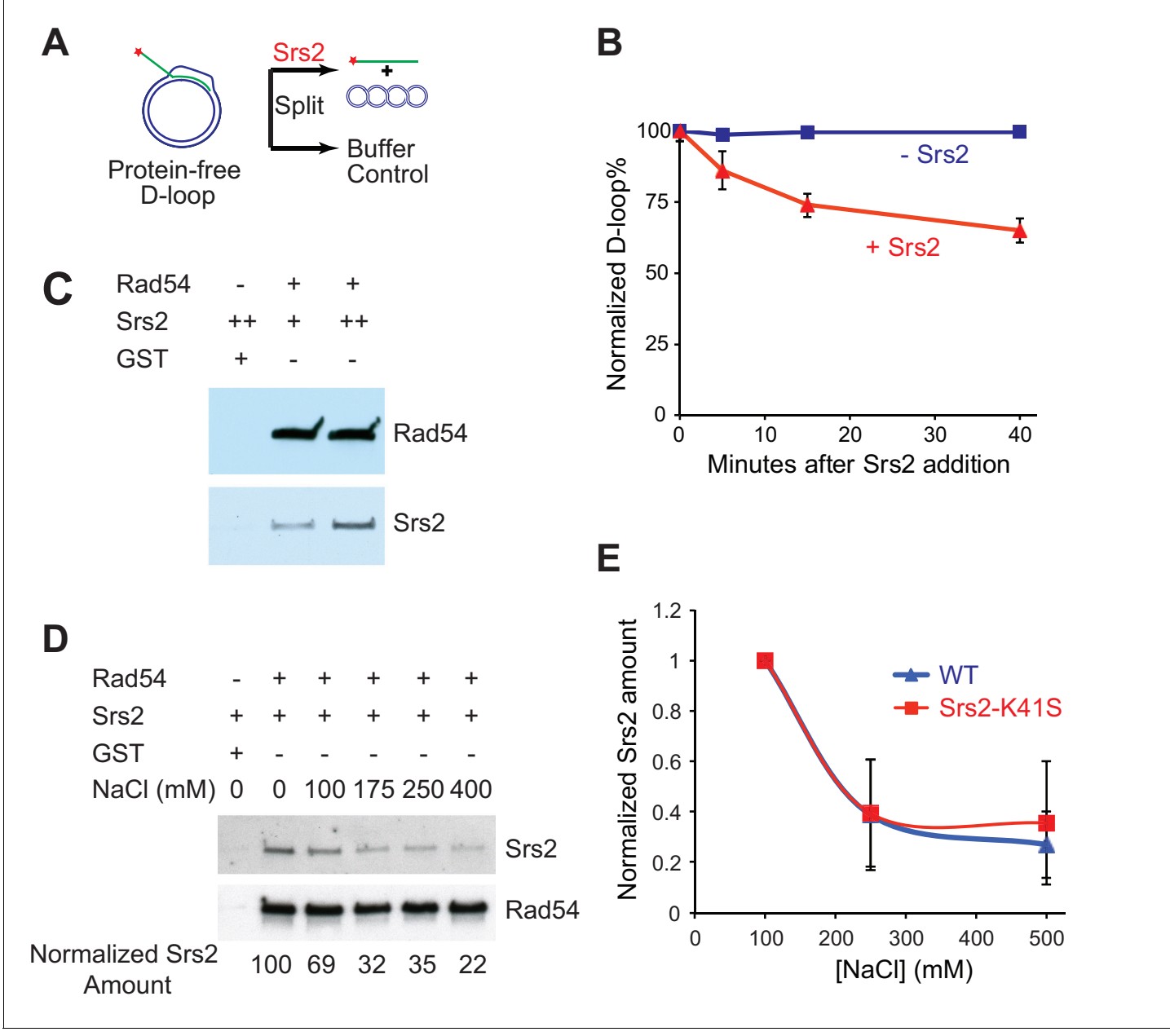

**Figure 3.** Srs2 disrupts Rad51/Rad54 reconstituted D-loops with higher efficiency than protein-free D-loops. (A) D-loop disruption assay using purified protein-free D-loops generated by Rad51 and Rad54. (B) Protein-free D-loops were split into buffer containing Srs2 or no protein and incubated at 30°C. Samples were taken at 0, 5, 15, and 40 min, and the final concentration of Srs2 is 15 nM. (C) GST-Rad54 physically interacts with Srs2. 17.5 nM GST-Rad54 or 175 nM GST were incubated with either 8.75 nM or 17.5 nM of Srs2 for 1 hr before pulldown. (D) The protein interaction between Rad54 and Srs2 is sensitive to increasing ionic strength. GST-Rad54 and Srs2 were formed in buffer containing 0, 100, 175, 250, or 400 mM NaCl before pulldown. (E) Srs2-K41S interacts with Rad54 with similar salt sensitivity, compared to wild type Srs2. Both wild type Srs2 and Srs2-K41S were allowed to form complex with GST-Rad54 in buffer containing 0, 250, and 500 mM NaCl before pulldown. Plotted are means ± standard deviation from n = 3. In (D) and (E), 31.3 nM GST-Rad54 or 313 nM GST (GE Healthcare) were incubated with 31.3 nM of Srs2 or Srs2-K41S in the same buffer containing indicated amount of NaCl for 1 hr before pulldown. Pulldown Srs2 amount in buffer containing 0 mM NaCl was normalized to 100 (D) or 1 (E).

The following source data is available for figure 3:

**Source data 1.** Source data for *Figure 3B*.
**Source data 2.** Source data for *Figure 3E*.

Rad51 from ssDNA (*Antony et al., 2009*). Only amino acids 783–859 of the Rad51 protein interaction site of Srs2 were found to be required for non-crossover formation in vivo but not amino acids 860–998 (*Miura et al., 2013*). In the absence of more defined Rad51 interaction mutations, it is difficult to assess whether Rad51 interaction is required for SDSA. Rad54 removes Rad51 at the 3' end of the invading strand during heteroduplex formation to clear the stage for DNA polymerases (*Li and Heyer, 2009*; *Wright and Heyer, 2014*), suggesting that Rad51 may not be in a position to interact with Srs2 during D-loop disruption. Therefore, we entertained the idea that Srs2 might interact with Rad54 and discovered a direct and robust physical interaction between Srs2 and Rad54 (*Figure 3C,D*). The Srs2-Rad54 interaction might be involved in the observed preference to disrupt D-loops in reconstituted reactions using the cognate proteins. We conclude that interactions with Rad51 and Rad54, but not RPA (see *Figure 1—figure supplement 1*), potentially play an important role in Srs2-mediated D-loop disruption.

## Srs2 prefers to disrupt D-loops where the 3'-end is part of the heteroduplex DNA

To determine whether Srs2 has a preferred polarity in D-loop disruption, we used a previously characterized hDNA digestion assay (*Wright and Heyer, 2014*). Formation and decrease of hDNA regions were assessed by the ability of the invading strand to be digested by restriction enzymes once the cut site has been made double-stranded by hDNA formation in the D-loop (*Figure 4A*). The 3 kb plasmid donor will allow incorporation of about 200 nucleotides of the invading strand into hDNA, limited by the number of negative supercoils (*Li et al., 2009*). For D-loops formed by the invasion of *607*, this means that the heteroduplex region will not contain the *Mlu*I and *Bst*XI sites (362 nucleotides apart) simultaneously. The observation that the sum of the *Mlu*I (5') and *Bst*XI (3') digestion products matches exactly the amount of D-loop products (*Figure 4C,D*) independently confirms that both restriction sites never cohabitate, otherwise a sum of greater than 100% of the D-loop amount would be expected. Thus, there were two major species of D-loop, *Mlu*I-incorporated D-loop (30% yield) and *Bst*XI-incorporated D-loop (42% yield) (*Figure 4A,E*). The addition of Srs2 to an ongoing Rad51/Rad54-reconstituted D-loop reaction revealed a marked preference for disruption of D-loops in which the 3' end-proximal region (*Bst*X1 cleavage is 111 nt from the 3' end) had been incorporated into hDNA. The 3' proximal/*Bst*XI-incorporated D-loops show an immediate and rapid decline from 42 to 10 percent in the ~30 s it took to add the enzyme and stop the reaction (*Figure 4E*). In contrast, the 5' proximal *Mlu*I -incorporated D-loops showed a slower time course of disruption that did not reach a final level until 15 min (*Figure 4E*). As is the case in the absence of Srs2, the D-loop levels closely follow the sum of 3' and 5' proximal digestion products, and the cleavage levels never decrease for one site while increasing for the other (*Figure 4D*), indicating that Srs2 action does not result in extensive branch migration of the hDNA region before complete hDNA disruption is accomplished. We conclude that Srs2 prefers disrupting D-loops, which include the 3' region of the invading DNA in the heteroduplex DNA. This is in congruence with the data of *Figure 2B–D*, where ds98-*607* D loops are preferentially disrupted by Srs2, followed by *607* and then gapped ssDNA. This follows the same rank order of 3' proximal site incorporation preference that was previously mapped by the hDNA incorporation assay for these same three substrates under identical conditions (*Wright and Heyer, 2014*). Further, this may reflect the need for Srs2 to disrupt 3' heteroduplex to promote the D-loop reversal step of SDSA.

## Srs2 disrupts D-loops extended by DNA polymerase δ in the presence of PCNA/RFC

The presumed D-loop substrate for Srs2 disruption during SDSA is the extended D-loop that is engaged by DNA polymerase δ with PCNA/RFC, as Pol δ is the major polymerase active in HR-associated first-end DNA synthesis extending the invading strand (*Li et al., 2009*; *McVey et al., 2016*; *Sebesta et al., 2011*). To present Srs2 with this substrate, we partly reconstituted the SDSA pathway in vitro using Rad51, Rad54, RPA, PCNA, RFC, and DNA polymerase δ and assessed the ability and efficiency of Srs2 in disrupting extending D-loops (*Figure 5*). The long substrates we tested in the earlier assays produce D-loops with about 200 base pairs of heteroduplex incorporation, resulting in fully relaxed plasmid donors. Consequently, extended D-loops do not differentiate from unextended D-loops during electrophoresis, and in addition, DNA polymerase δ would experience an immediate

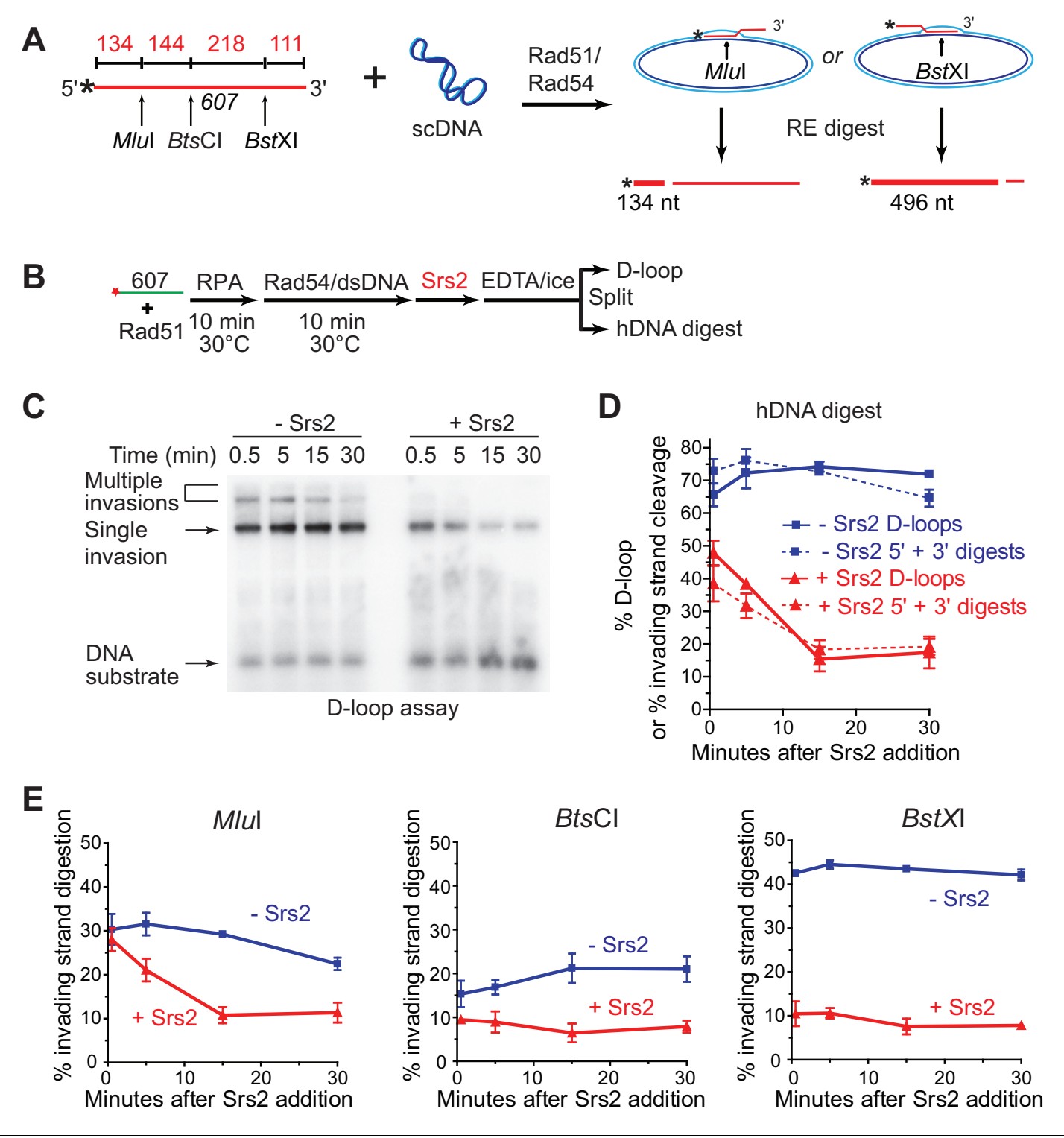

**Figure 4.** Srs2 preferentially disrupts D-loops with 3' proximal hDNA. (A) hDNA detection assay. Note that since the *Bst*XI and *Mlu*I sites are 362 nt apart, they cannot be incorporated into the same plasmid DNA molecule in a contiguous hDNA tract, and therefore the two sites are mutually exclusive and report on the formation of D-loop products with 3' or 5' proximal hDNA content. (B) Scheme of the experiment. (C) Representative D-loop assay gel. Times points are post-Srs2 (5 nM) or dilution buffer control addition, starting 10 min after the initiation of the D-loop reaction with Rad54 protein and homologous donor plasmid addition. (D) Quantitation of D-loops and the sum of the 3' and 5' proximal digestion products. (E) Quantitation of hDNA digestion at three sites on the invading DNA molecule, as indicated in the schematic at top. Note that the middle *Bts*CI site is close enough to

*Figure 4 continued on next page*

*Figure 4 continued*

the two other sites that it is not mutually exclusive for incorporation into a fraction of the 5' or 3' proximal hDNA products. Plotted are means ± standard deviation from n = 3.

The following source data is available for figure 4:

**Source data 1.** Source data for *Figure 4*.

topological block to DNA synthesis when attempting to extend the hDNA region. Therefore, a conventional 100-mer ssDNA was used to allow clear visual difference between extended and unextended D-loops (see *Figure 5B*). The reaction was staged to recapitulate the flow of the SDSA pathway in vivo to maximize D-loop synthesis before the addition of Srs2 (*Figure 5A*). Rad51 and Rad54 were allowed to produce high levels of D-loop before the addition of RFC/PCNA and Pol δ. Srs2 was added after efficient D-loop extension. To minimize fluctuations, the newly extended D-loops were split at the end into buffers containing various amount of Srs2 to reach a final concentration of 0, 2.5, 7.5, or 25 nM, and products analyzed (*Figure 5A*). At the 25 min time point, *i.e.* 40 min after Pol δ addition, almost all D-loops had been extended and the yield of extended D-loop plateaued at 30% (*Figure 5B,C*). Addition of Srs2 led to a significant decrease in extended D-loops compared to the control without Srs2 (*Figure 5C*). 25 nM Srs2 decreases extended D-loop level from 30% to 7.6%, and correspondingly increases collapsed extended D-loops (*Figure 5C*). These data demonstrate that Srs2 dissociates Polδ-extending D-loops.

To corroborate this important conclusion, we devised an independent experimental design using $\alpha$-$^{32}$P-dCTP incorporation to specifically radiolabel D-loops upon extension by Polδ (*Figure 5D*). Using 25 nM of Srs2 (25 nM) essentially all extended D-loops were disrupted within the first five minutes, (*Figure 5D,E*, *Figure 5—figure supplement 1*). We conclude that Srs2 is capable of disrupting extended D-loops in the presence of RFC, PCNA, and Polδ, providing a potential mechanism for how Srs2 promotes SDSA and favors non-crossover formation. We estimate that disruption of RFC/PCNA/ Pol δ extending D-loop (*Figure 5C*) is at least twice faster than disruption of D-loops with the *607*-based invading DNA (*Figure 1C*). Thus, replication machinery-covered D-loops serve as better substrates for Srs2, suggesting a positive recruitment of Srs2 through protein-protein interaction, which we examined in more below.

## Srs2 exhibits slight preference for extending D-loop using SUMO-PCNA

Previous studies have established that SUMO-PCNA exhibits higher affinity to Srs2 than unmodified or ubiquitinated PCNA (*Papouli et al., 2005*; *Pfander et al., 2005*), leading to the model that SUMO-PCNA recruits Srs2 to its site of action. Genetic analyses showed that the protein interaction of Srs2 with SUMO-PCNA is required for SDSA in vivo, as *siz1△* and *pol30-K164R* mutant strains shared the same defects as *srs2△* strain in SDSA (*Le Breton et al., 2008*; *Miura et al., 2013*). This suggests that SUMO-PCNA and its interaction with Srs2 is required for SDSA and predicts that posttranslational modifications of PCNA affect the efficiency of Srs2 in D-loop disruption. Hence, we evaluated the effect of PCNA modification by SUMO and ubiquitin on Srs2-mediated D-loop disruption of Pol δ extending D-loops. Split forms of wild type PCNA (referred as WT PCNA thereafter), monoubiquitylated-PCNA (Ubi-PCNA), and SUMOylated-PCNA (SUMO-PCNA) were utilized, where either ubiquitin or SUMO was fused in frame to the C-terminal peptide to produce large quantities of Ubi-PCNA and SUMO-PCNA with uniform modifications (*Freudenthal et al., 2011*, *2010*). Crystal structures of split form Ubi-PCNA and SUMO-PCNA revealed that ubiquitin and SUMO moieties were localized on the back side of the PCNA trimeric ring without changing its conformation, and indeed these split forms of PCNA function properly at both the cellular and biochemical level (*Freudenthal et al., 2011*, *2010*). We reconstituted the in vitro SDSA assay with WT PCNA, Ubi-PCNA, and SUMO-PCNA separately and monitored whether Srs2 D-loop disruption activity is affected. These split forms of PCNA have been reported to support normal cell replication in yeast and stimulate the ability of pol η or Pol δ to incorporate nucleotide opposite an abasic site (*Freudenthal et al., 2011*, *2010*). Consistent with these reports, we did not detect a difference among different species of PCNA in recruiting Pol δ to nascent D-loops for new DNA synthesis and D-loop extension in the absence of Srs2 (*Figure 5E*, *Figure 6*, *Figure 6—figure supplement 1*,

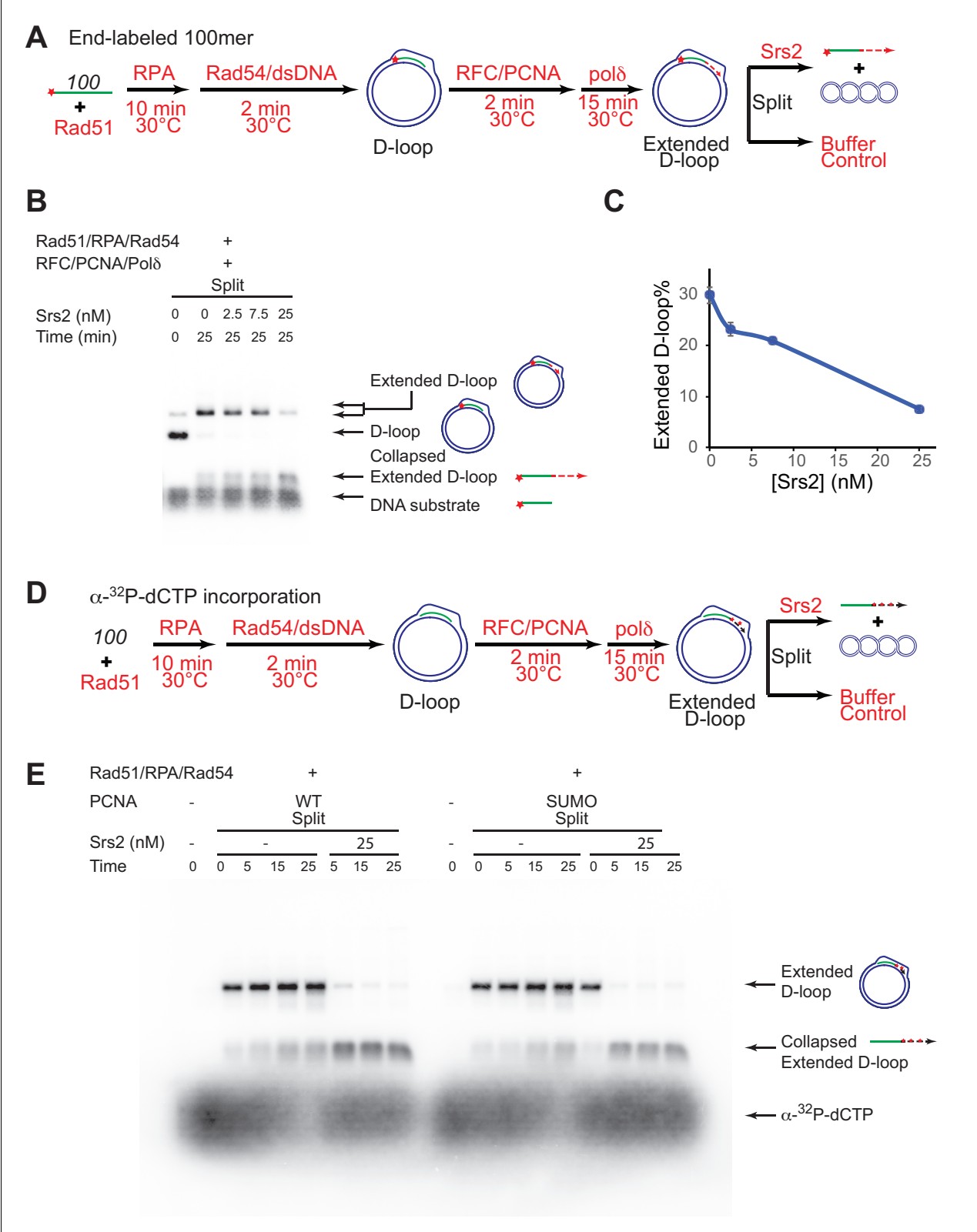

**Figure 5.** Srs2 disrupts extended D-loops produced by DNA polymerase δ, in a reconstituted assay system containing RFC, PCNA, Rad51 and Rad54. (**A**) Extended D-loop disruption assay. Fully homologous 100mer ssDNA was used as substrates. 0.2 μM Rad51 was incubated with 6 nM ssDNA substrates (Rad51/nt = 1:3) for 10 min at 30°C. RPA (165 nM) was added for another 10 min incubation before the addition of 84 nM Rad54 and 7 nM supercoiled plasmid dsDNA (21 μM bp). After 2 min, 20 nM RFC and 20 nM PCNA were added and incubated for another two min at 30°C before the

*Figure 5 continued on next page*

*Figure 5 continued*

addition of 7 nM DNA polymerase δ. After 15 min, different amount of Srs2 or storage buffer were added and samples at 25 min were taken and stopped by SDS/Proteinase K treatment. (**B**) Titration of Srs2 (0, 2.5, 7.5, or 25 nM final concentration) in extended D-loop disruption. (**C**) Quantitation of extended D-loops in **B**). Plotted are means ± standard deviation from n = 3. (**D**) Extended D-loop disruption assay using α-$^{32}$P-dCTP incorporation. Experimental setup and protein concentrations are identical as in **A**), except that α-$^{32}$P-dCTP was included in the buffer and 100mer ssDNA was not radiolabeled. (**E**) Time course of extended D-loop disruption by Srs2, in the presence of PCNA and SUMO-PCNA. After 15 min, Srs2 (25 nM final concentration) or storage buffer were added and samples at 0, 5, 15, and 25 min were taken and stopped by SDS/Proteinase K treatment.

The following source data and figure supplements are available for figure 5:

**Source data 1.** Source data for *Figure 5C*.
**Figure supplement 1.** Srs2 disrupts WT PCNA and SUMO-PCNA extending D-loops with similar kinetics at optimal conditions.
**Figure supplement 1—source data 1.** Source data for *Figure 5—figure supplement 1*.

*Figure 7—figure supplement 1*). Initial testing was set up as in *Figure 5A and D*, where Pol δ extension time and reaction temperature are optimized to maximize extended D-loop. No difference was observed by high concentration (25 nM) of Srs2 in disrupting extended D-loop with WT PCNA or SUMO-PCNA (*Figure 5E*, *Figure 5—figure supplement 1*).

To detect a potential preference for a form of PCNA (SUMO, ubiquitin, unmodified) we moved away from optimized reaction conditions. As shown in *Figure 6A*, we lowered the temperature from 30°C to 25°C to slow down RFC/PCNA/Pol δ, shortened the Pol δ extension time to avoid accumulating fully extended D-loops, and titrated a lower range of Srs2 concentrations. The results show a slight, but significant, preference to disrupt SUMO-PCNA extending D-loops over WT PCNA- or Ubi-PCNA extending D-loops at a limiting concentration of Srs2 (7.5 nM) (*Figure 6*). High concentrations of Srs2 (25 nM) disrupt extended D-loops with all species of PCNA equally well in this experimental setup (*Figure 6—figure supplement 1*). To obtain more precise quantitation, we repeated this reaction scheme with 7.5 nM Srs2 using end-labeled invading DNA substrate. As shown in *Figure 7*, *Figure 7—figure supplement 1*, the total D-loop levels were almost identical across different species of PCNA, all showing a decline from ~42% to ~32% after 25 min incubation with Srs2, compared to an increase from ~40% to ~48% in total D-loops without Srs2. Importantly, the level of extended D-loops was lower while the unextended D-loops were, if anything, higher when comparing SUMO-PCNA *vs*. WT PCNA or Ubi-PCNA in the presence of Srs2 (*Figure 7D,E*). The ratio of extended D-loops over unextended D-loops reflects this difference, revealing that Srs2 prefers to disrupt extended D-loops over unextended D-loops when SUMO-PCNA is present (*Figure 7F*). When D-loop extension was prevented by the omission of dNTPs, no significant difference in D-loop disruption by Srs2 was noted between reactions containing PCNA, Ubi-PCNA, or SUMO-PCNA (*Figure 7—figure supplement 2*).

To test the idea whether PCNA actively recruits Srs2 onto newly extended D-loops, we also compared the Srs2 disruption efficiency when the replication machinery was only loaded at the nascent D-loops without new DNA synthesis because of the absence of dNTP to the condition when no replication machinery was loaded. We only detected a slight difference between the conditions when SUMO-PCNA/RFC/Pol δ was present or not (*Figure 7—figure supplement 3*). From these experiments we conclude, that Srs2 preferentially disrupts D-loops that are actively extending by DNA Polymerase δ using SUMO-PCNA.

## Discussion

Genetic observations that Srs2 suppresses crossover formation and promotes the SDSA pathway have suggested a biochemical role for Srs2 to disrupt D-loops and to channel the recombination intermediates into the SDSA non-crossover pathway (*Dupaigne et al., 2008*; *Ira et al., 2003*; *Le Breton et al., 2008*; *Robert et al., 2006*). Early biochemical studies on Srs2 either did not employ the D-loop reaction or staged the reaction to focus on the anti-recombination activity of Srs2 (*Krejci et al., 2003*; *Veaute et al., 2003*). Several later studies failed to observe D-loop

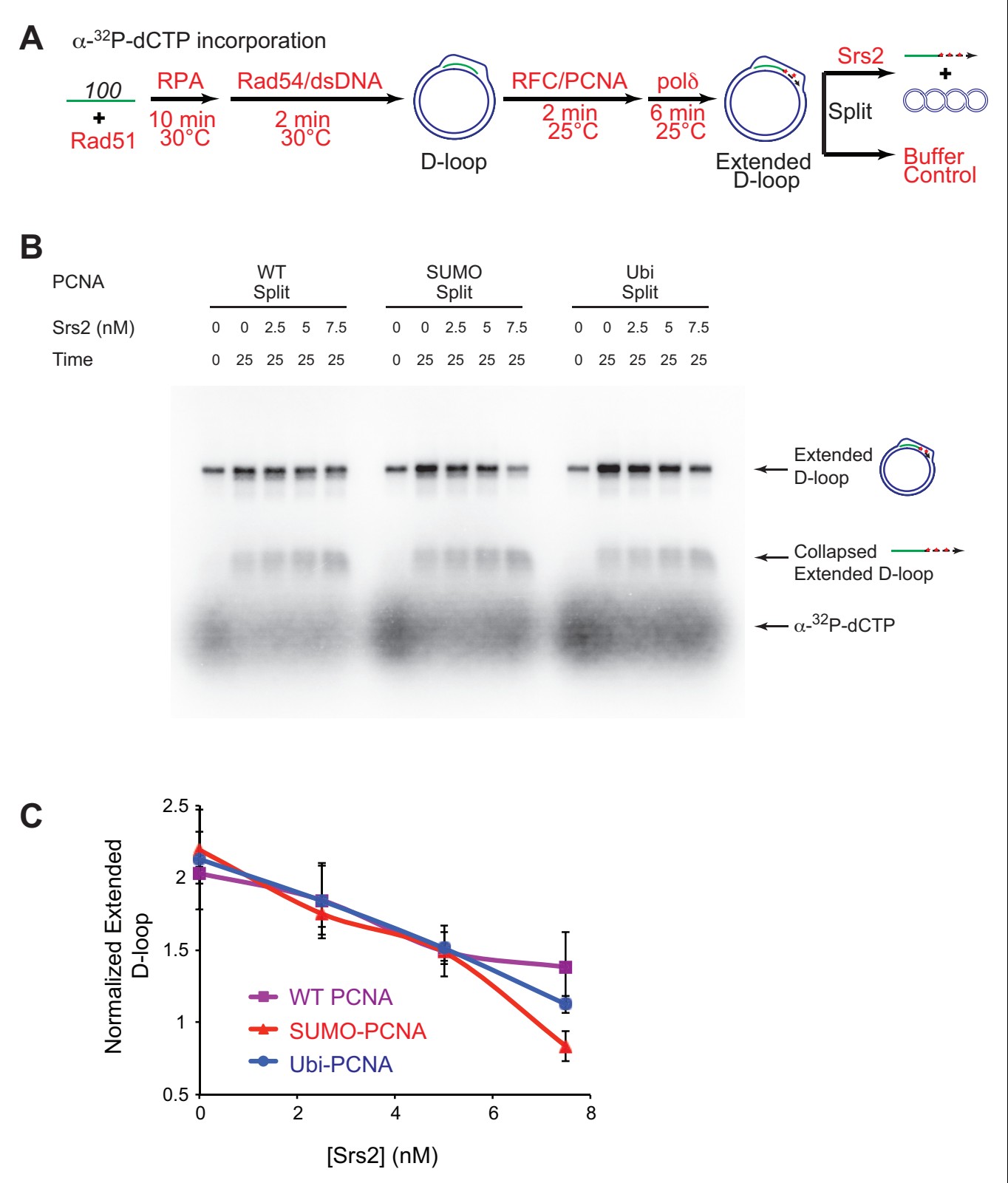

**Figure 6.** Slight preference of Srs2-mediated disruption for PCNA-SUMO containing reconstituted reactions compared to unmodified or ubiquitinated PCNA. (**A**) Extended D-loop disruption assay using α-$^{32}$P-dCTP incorporation. Fully unlabeled and homologous 100mer ssDNA was used as substrates, and α-$^{32}$P-dCTP was included in the buffer. 0.2 μM Rad51 was incubated with 6 nM ssDNA substrates (Rad51/nt = 1:3) for 10 min at 30°C. RPA (165 nM) was added for another 10 min incubation before the addition of 84 nM Rad54 and 7 nM supercoiled plasmid dsDNA (21 μM bp). After 2 min, 20 nM
*Figure 6 continued on next page*

*Figure 6 continued*

RFC and 20 nM PCNA, PCNA-SUMO, or PCNA-Ubi, were added and temperature was lowered to 25°C. DNA polymerase δ (7 nM) was added after two more min and incubated for 6 min at 25°C. Reactions with PCNA, SUMO-PCNA, and Ubi-PCNA were individually split into Srs2 or buffer only control. Final concentrations of Srs2 are 0, 2.5, 7.5, or 25 nM. Samples at 25 min were taken and stopped by SDS/Proteinase K treatment. (B) Titration of Srs2 in extended D-loop disruption. (C) Quantitation of normalized extended D-loops in (B). The intensities of the extended D-loops were normalized against the signal of the extended D-loop produced with wild type PCNA without Srs2 at 0 min in each repeat. Plotted are means ± standard deviation from n = 3.

The following source data and figure supplement are available for figure 6:

**Source data 1.** Source data for *Figure 6C*.

**Figure supplement 1.** High concentrations of Srs2 are equally proficient to disrupt D-loops extending with different forms of PCNA.

disruption activity by Srs2 in reconstituted in vitro reactions although using significantly higher concentrations of Srs2 (up to 167 nM) than used here (*Prakash et al., 2009*; *Sebesta et al., 2011*). We have identified several factors that explain this apparent discrepancy of the results reported here with these studies. First, the traditional short oligonucleotide-based D-loops undergo significant instability caused by Rad54 (*Bugreev et al., 2007a*; *Wright and Heyer, 2014*), making it difficult to identify an effect of Srs2 on D-loop stability. Moreover, the effect of Srs2 in disrupting D-loops becomes more apparent at later time points of Srs2 incubation, in particular in the reconstituted reactions with PCNA/RFC and Polδ (*Figures 1*, *5–7*, *Figure 7—figure supplements 2* and *3*). These differences in protocol explain why no disruption was observed when Srs2 was added 1 min after plasmid dsDNA addition and the reaction time was only 8 min (*Prakash et al., 2009*) or only 5 min (*Sebesta et al., 2011*). The D-loop disassembling activity of Srs2 was much more evident, when we used D-loops made of long ssDNA with physiological-relevant length and structure, which minimize Rad54-mediated D-loop instability (*Figures 1* and *2*). Another previous study showed that the Srs2 C-terminal truncation containing amino acids 1–898 was able to disrupt protein-free D-loops assembled from oligonucleotides (*Marini and Krejci, 2012*). This result showed that Srs2 can disrupt model D-loops, but left open the critical question, whether Srs2 can act in the context of the ongoing reaction, where proteins are bound to the HR-intermediates, in particular in extended D-loops which engage PCNA/RFC and DNA Polymerase δ. For example, Sgs1 disrupts protein-free-D-loops, but Sgs1 is not able to disrupt D-loops in reconstituted reactions (*Fasching et al., 2015*). Importantly, we were able to partly reconstitute the SDSA pathway and show that Srs2 efficiently disrupts D-loops in an ATP-dependent fashion (*Figures 1* and *5*). These results are consistent with the genetic requirement for the Srs2 ATPase activity for SDSA (*Kolesar et al., 2016*; *Miura et al., 2013*). These findings provide a plausible mechanism for the SDSA function of Srs2 as discussed below (*Figure 8*).

## Mechanisms of synthesis-dependent strand annealing: A model for Srs2

SDSA appears to be the primary HR pathway in somatic cells and includes a critical step of dissociating the extended D-loop to allow annealing of the newly synthesized strand with the processed second end of the DSB (*Figure 8*). Mph1 has been implicated in this step by genetic and biochemical experiments (*Prakash et al., 2009*; *Sebesta et al., 2011*) and genetic experiments strongly suggested that other enzymes, specifically Srs2, play a partially overlapping role (*Mitchel et al., 2013*; *Prakash et al., 2009*). Concomitant with DNA strand invasion by the Rad51 filament, Rad54 is required to remove Rad51 during hDNA formation to allow access to PCNA loading by RFC to enable DNA synthesis by DNA Polymerase δ (*Li et al., 2013*, *2009*). We speculate that sumoylation of PCNA during HR-associated DNA synthesis increases the likelihood of Srs2 recruitment to the extending D-loop, where it disengages the polymerase from PCNA, as Srs2 has been shown to compete with Pol δ for PCNA binding (*Burkovics et al., 2013*). Srs2 contains a PCNA interaction motif (PIP box) and a SUMO interaction motif (SIM) providing the basis for the avid interaction between Srs2 and SUMO-PCNA (*Armstrong et al., 2012*; *Papouli et al., 2005*; *Pfander et al., 2005*). This enhanced recognition between Srs2 and SUMO-PCNA has classically been viewed as an effective means to eliminate inappropriate recombination events through Rad51 filament disruption

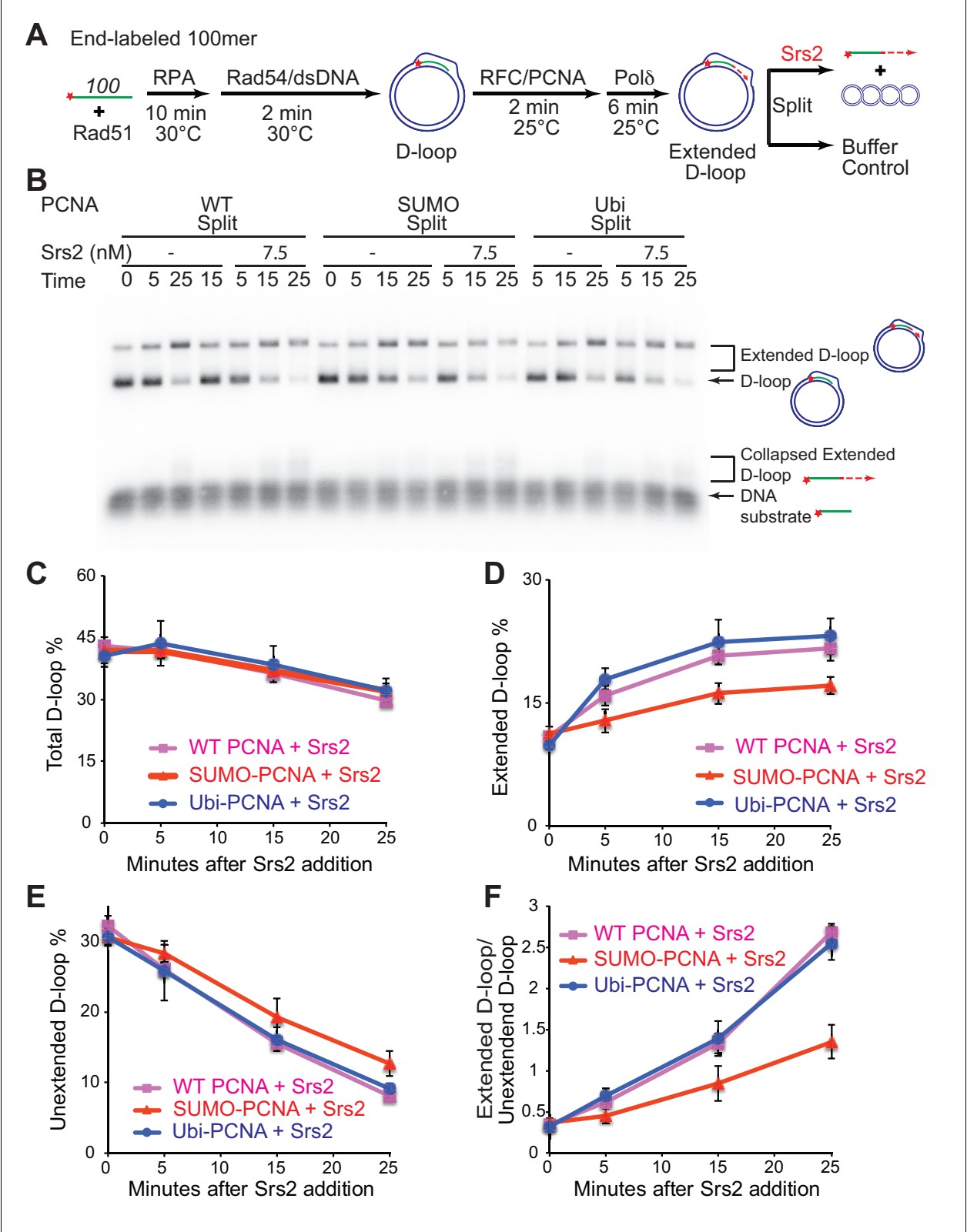

**Figure 7.** Srs2 prefers to disrupt extended D-loop over unextended D-loop when SUMO-PCNA is present in a reconstituted system compared to unmodified or Ubi PCNA. (**A**) Extended D-loop disruption assay. Fully homologous 100mer ssDNA was used as substrates. Rad51 (0.2 μM) was incubated with 6 nM ssDNA substrates (Rad51/nt = 1:3) for 10 min at 30°C. RPA (165 nM) was added for another 10 min incubation before the addition of 84 nM Rad54 and 7 nM supercoiled plasmid dsDNA (21 μM bp). After 2 min, 20 nM RFC and 20 nM PCNA, SUMO-PCNA, or Ubi-PCNA, were added

*Figure 7 continued on next page*

*Figure 7 continued*

and temperature was lowered to 25°C. DNA polymerase δ (7 nM) was added after two more min and incubated for 6 min at 25°C. Reactions with PCNA, SUMO-PCNA, and Ubi-PCNA were individually split into Srs2 or buffer only control. (B) Time course of extended D-loop disruption by 7.5 nM Srs2. Samples were taken at 0, 5, 15, and 25 min and stopped by SDS/Proteinase K treatment. (C) Quantitation of total D-loops in (C), unextended D-loops in (D), extended D-loops in (E), and ratio of extended D-loops/unextended D-loops in (F). Plotted are means ± standard deviation from n = 3.

The following source data and figure supplements are available for figure 7:

**Source data 1.** Source data for *Figure 7* and *Figure 7—figure supplement 1*.

**Figure supplement 1.** Different species of PCNA, unmodified PCNA, SUMO-PCNA, and Ubi-PCNA, have no impact on the level of unextended D-loops and extended D-loops in the absence of Srs2.

**Figure supplement 2.** Different species of PCNA, unmodified PCNA, SUMO-PCNA, and Ubi-PCNA, have no impact on the level of D-loops when D-loop extension is blocked.

**Figure supplement 2—source data 1.** Source data for *Figure 7—figure supplement 2*.

**Figure supplement 3.** The presence of SUMO-PCNA at the nascent D-loop during Srs2-mediated D-loop disruption.

**Figure supplement 3—source data 1.** Source data for *Figure 7—figure supplement 3*.

(*Papouli et al., 2005*; *Pfander et al., 2005*; *Stelter and Ulrich, 2003*). Based on the biochemical preference of Srs2 to disrupt D-loops that are extended using SUMO-PCNA (*Figures 6* and *7*), we suggest that PCNA sumoylation is also critical for the tuning of the SDSA function of Srs2. This is consistent with the genetic data showing a requirement for the ATPase, the PIP box and the SIM for the pro-SDSA function of Srs2 (*Burkovics et al., 2013*; *Kolesar et al., 2016*; *Le Breton et al., 2008*; *Miura et al., 2013*; *Robert et al., 2006*). Additionally, ablation of Siz1-mediated sumoylation of PCNA restores the formation of damage-induced and Rad51-dependent X-shaped sister chromatid junctions in an *sgs1△ rad18△* strain background (*Branzei et al., 2008*), which suggests a positive role of SUMO-PCNA in processing such HR intermediates. It is possible, but remains to be demonstrated, that SUMO-PCNA stays associated with the D-loop and Srs2 during D-loop disruption (see *Figure 8*). The orientation of Srs2 in disrupting the D-loop in *Figure 8* by translocating 3′−5′ on the invading strand is consistent with the known polarity of Srs2 (*Rong and Klein, 1993*) and the preference of Srs2 for D-loops with an embedded 3′-end (*Figures 2* and *4*).

Srs2 exhibits a significant preference for disrupting D-loops in the cognate reconstituted reactions compared to protein-free DNA intermediates (*Figures 1* and *3*), suggesting that this activity is of biological relevance and likely related to the physical interactions of Srs2 with Rad51 (*Colavito et al., 2009*; *Seong et al., 2009*) and/or Rad54 (*Figure 3*). Indeed, Srs2 stimulates ATP hydrolysis activity of Rad51 and triggers the dissociation of Rad51 clusters from ssDNA (*Antony et al., 2009*). The same principle could be utilized to dislodge residual Rad51 and Rad54 from newly extended D-loops, which could further destabilize hDNA by removing these potentially stabilizing protein factors. It was reported that Rad51 bound to adjacent duplex DNA stimulates the helicase activity of Srs2 (*Dupaigne et al., 2008*).

As depicted in *Figure 8* and proposed for Srs2 and RECQ5 (*Mitchel et al., 2013*; *Schwendener et al., 2010*), it is possible that Srs2 also acts on the second end during SDSA dissociating Rad51 to enable reannealing, as Rad51 inhibits the Rad52 annealing reaction (*Wu et al., 2008*). This mechanism would be complementary to Srs2 disrupting D-loops, as Rad51 displacement alone appears to be insufficient to explain the SDSA role of Srs2. Analysis of the *srs2-R1* mutant showed that the interaction of Srs2 with SUMO-PCNA is not required for its anti-recombination activity but the mutant showed the increase in crossovers indicative of a defect in SDSA (*Le Breton et al., 2008*). In fact, the resected DSB is not a substrate for PCNA loading by RFC, as it has the opposite polarity (5′ junction) of the native RFC substrate (3′-junction) (*Majka and Burgers, 2004*), and it appears unlikely that PCNA would be found on resected DSBs.

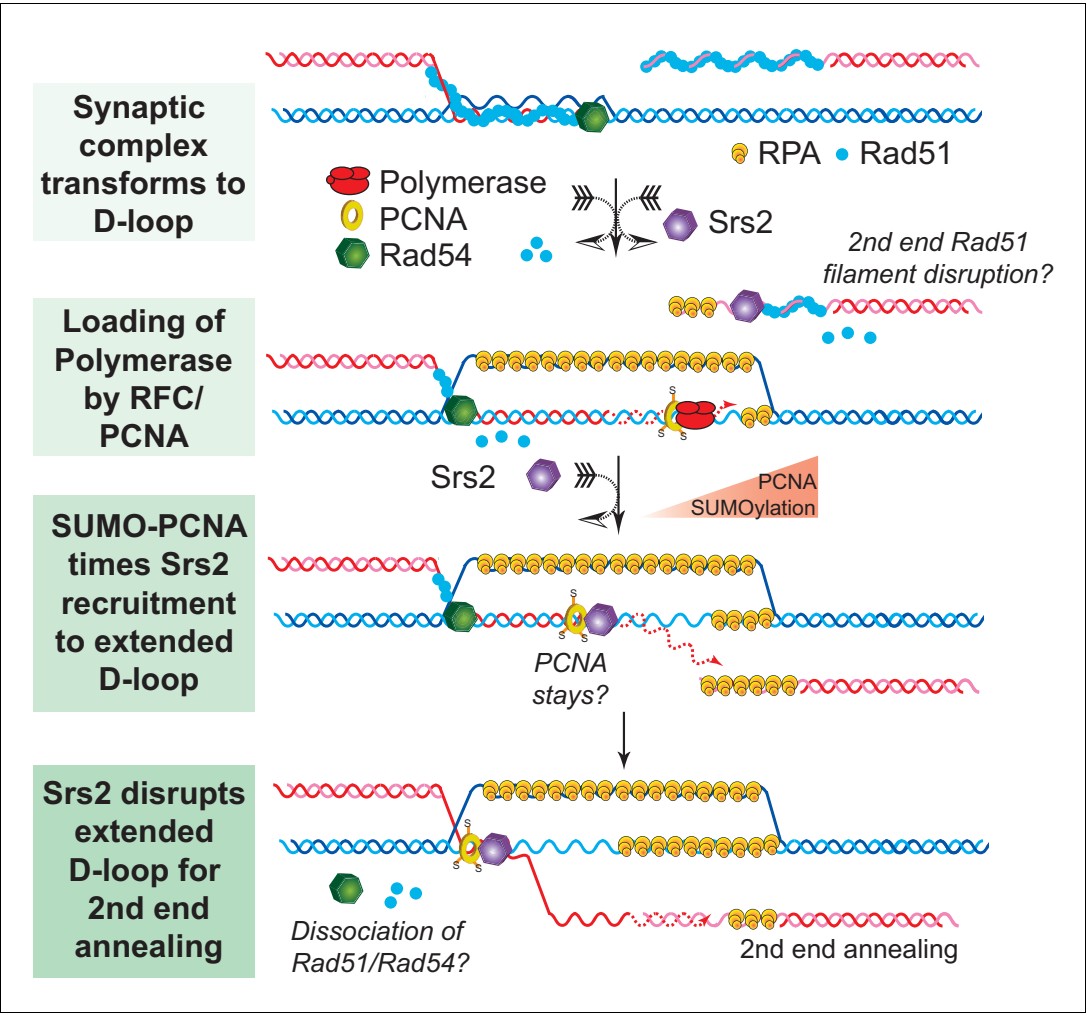

**Figure 8.** SDSA Model: SUMO-PCNA facilitates Srs2 recruitment towards newly extended D-loops. After DNA resection and Rad51 filament formation, Rad54 assists Rad51 to search for homology and form D-loops. During heteroduplex formation, Rad54 removes Rad51 at the 3' invading end to prepare the loading of DNA polymerase through RFC and PCNA. DNA synthesis is initiated by DNA polymerase δ. SUMO-PCNA actively recruits Srs2 to the D-loop. Srs2 potentially enforces SDSA through three distinct mechanisms: 1) Srs2 competes with DNA polymerase for PCNA controlling the length of new DNA synthesis (*Burkovics et al., 2013*). 2) Srs2 translocates in a 3' → 5' direction to dissociate the hDNA incorporated in the D-loop (this work). After the newly extended 3' invading end is dissociated it is available to anneal with the second end to generate non-crossover products. 3) Srs2 may also dissociate Rad51 from the second end to allow single-strand annealing in analogy to RECQ5 (*Mitchel et al., 2013*; *Schwendener et al., 2010*).

The following figure supplement is available for figure 8:

**Figure supplement 1.** Why are so many proteins involved in D-loop disruption? Different D-loops during homologous recombination initiating at DNA double-stranded breaks and replication-associated gaps.

In sum, Srs2 and Mph1 appear to provide two distinct pathways to process extended D-loops and shunt HR to non-crossover outcome and SDSA. The genetic profiles of both mutants show a clear difference: *mph1* mutants shift non-crossover to crossover events without effect on repair efficiency, whereas *srs2* mutants show a decrease in repair efficiency and loss of non-crossover products (*Mitchel et al., 2013*; *Prakash et al., 2009*; *Ira et al., 2003*). The synthetic negative interaction between both mutants suggests partial overlap between both pathways (see *Figure 8—figure supplement 1*) for a discussion why there are so many proteins involved in D-loop disruption). What

differentiates both proteins is their physical and functional interaction with PCNA, specifically SUMO-PCNA. We suggest that Srs2 is specialized to disrupt D-loops extended with SUMO-PCNA, and that in its absence such extended D-loops become a potentially lethal intermediate.

## Material and methods

### Proteins and DNA substrates

Yeast Rad51, RPA, Rad54, Srs2, RFC, and DNA Polymerase δ were purified as described (*Fortune et al., 2006*; *Garg et al., 2005*; *Kiianitsa et al., 2002*; *Papouli et al., 2005*; *Solinger and Heyer, 2001*; *Van Komen et al., 2006*). PCNA, Ubi-PCNA, and SUMO-PCNA in split forms were purified as published (*Freudenthal et al., 2011*, *2010*). Long ssDNA substrates with full homology and terminal heterologies and plasmid dsDNA donor for D-loop assay were prepared as described (*Wright and Heyer, 2014*).

### D-loop disruption assay

In short, Rad51-cataylzed D-loops were made before the addition of Srs2. First, Rad51 (0.2 μM) was incubated with various $^{32}$P-labeled ssDNA substrates (1 nM molecule, 0.61 μM in nucleotides, or 0.61–0.78 μM in nucleotides/base pairs) for 10 min at 30°C in a D-loop buffer containing 35 mM Tris-acetate pH 7.5, 100 mM NaCl, 7 mM magnesium-acetate, 2 mM ATP, 1 mM DTT, 0.25 mg/mL BSA, 20 mM phosphocreatine and 100 μg/mL phosphocreatine kinase. RPA (33 nM) was then added and incubated for another 10 min at 30°C. Rad54 (84 nM) and supercoiled plasmid dsDNA (7 nM molecule, 21 μM bp) were added and incubated for another 10 min at 30°C. Indicated amounts of Srs2 were added and incubated at 30°C. Equal volume of enzyme mixture with different amount of Srs2 and storage buffer was added to the reaction to deliver the final concentration of Srs2 as indicated. Samples were taken out at indicated time points, deproteinized, and separated on 1% agarose gels. Gels were dried and analyzed by a Storm phosphorimager. All the bands were quantified through densitometry using ImageQuant.

### Protein-free D-loop disruption assay

Rad51/Rad54-catalyzed D-loops were prepared using *607* substrate as described above and further purified with a G25 spin column to remove SDS and protease K. The yield was about 78% as determined by densitometry. The protein-free D-loop (0.78 nM *607* substrate in total) was incubated with 15 nM Srs2 or buffer control in the same D-loop reaction buffer containing 35 mM Tris-acetate pH 7.5, 100 mM NaCl, 7 mM magnesium-acetate, 2 mM ATP, 1 mM DTT, 0.25 mg/mL BSA, 20 mM phosphocreatine and 100 μg/mL phosphocreatine kinase. Samples were taken and analyzed as describe above.

### Extended D-loop disruption assay

Concentrations of different proteins and incubating temperature are indicated in figures and texts. In general, Rad51 (0.2 μM) was incubated with $^{32}$P-labeled 100mer substrate (6.1 nM molecule, 0.61 μM in nucleotides) for 10 min at 30°C in a D-loop buffer containing 35 mM Tris-acetate pH 7.5, 100 mM NaCl, 7 mM magnesium-acetate, 2 mM ATP, 1 mM DTT, 0.25 mg/mL BSA, 20 mM phosphocreatine, 100 μg/mL phosphocreatine kinase, and 100 μM each dATP, dGTP, dCTP, and dTTP. RPA (165 nM) was then added and incubated for another 10 min at 30°C. Rad54 (84 nM) and supercoiled plasmid dsDNA (7 nM molecule, 21 μM bp) were added and incubated for another 2 min at either 30°C or 25°C, as specified. RFC (20 nM) and PCNA (20 nM trimeric form) were added and incubated for another 2 min at either 30°C or 25°C, as specified. Pol δ (7 nM) was added and incubated for another 6 or 15 min at either 30°C or 25°C, as specified. Srs2 (0, 2.5, 5, 7.5 or 25 nM) was added and incubated at either 30°C or 25°C, as specified. For α-$^{32}$P-dCTP incorporation, 100 μM each of dATP, dGTP, dTTP, and only 0.22 μM of dCTP, spiked with α-$^{32}$P-dCTP, were included in the reaction buffer instead. Samples were taken out at the indicated time points, deproteinized, and separated on 1% agarose gels. Gels were dried and analyzed by a Storm phosphorimager. All the bands were quantified through densitometry using ImageQuant.

**Table 1.** *Saccharomyces cerevisiae* strains used in this study.

| WT | ura3::2x(URA3 Ylplac211 kankanMX4 URA3) | (Ede et al., 2011) |
|---|---|---|
| srs2Δ | ura3::2x(URA3 Ylplac211 kankanMX4 URA3)srs2Δ::HisMX6 | This Study |
| srs2-K41A | ura3::2x(URA3 Ylplac211 kankanMX4 URA3)srs2Δ::HisMX6 LEU2::srs2-K41A | This Study |
| WDHY 3429 | Wild-type | This Study |
| WDHY 3514 | rad18::LEU2 | Dr. Giordano Liberi (CY8564) |
| WDHY 3858 | srs2Δ::KanMX | This Study |
| WDHY 4083 | rad18::LEU2 srs2Δ::KanMX | This Study |
| WDHY 3851 | srs2-K41A | Dr. Hanna Klein |
| WDHY 4082 | rad18::LEU2 srs2-K41A | This Study |
| WDHY 4130 | srs2-K41R | Dr. Hanna Klein |
| WDHY 4141 | rad18::LEU2 srs2-K41R | This Study |
| WDHY 5473 | ura3::2x(URA3 Ylplac211 kankanMX4 URA3) | This Study |
| WDHY 5474 | ura3::2x(URA3 Ylplac211 kankanMX4 URA3) rad51::LEU2 | This Study |
| WDHY 5475 | ura3::2x(URA3 Ylplac211 kankanMX4 URA3) srs2::hisMX6 | This Study |
| WDHY 5476 | ura3::2x(URA3 Ylplac211 kankanMX4 URA3) rad51::LEU2 srs2::hisMX6 | This Study |

All WDHY strains listed are derivatives of W303 with the common genotype *MATa ade2-1 can1-100 his3-11,15 leu2-3,112 trp1-1 ura3-1 RAD5* (*Zhao et al., 1998*). All the other strains are derivatives of CEN.PK2-1C (*Ede et al., 2011*).

## Heteroduplex digestion assay

Heteroduplex digestion assay was carried out as published (*Wright and Heyer, 2014*). D-loop reactions were assembled as described above, but samples at different time points were stopped with EDTA only but without SDS and proteinase K. D loop reactions were carried out in 30 µl volumes as indicated in the experimental scheme (*Figure 4B*). At the indicated time points, 6 µl of the reactions were stopped with EDTA delivered in 0.75 µl to a final concentration of 20 mM and placed on ice. 1.5 µl of each stopped time point was processed for agarose gel visualization of D-loop products or digested individually with each of the indicated restriction enzymes and products visualized by denaturing PAGE analysis as described (*Wright and Heyer, 2014*).

## Pulldown assay

The GST-tag based protein pulldown assay was carried out as previously described (*Liu et al., 2011*). GST-Rad54 or GST (GE Healthcare) were incubated with Srs2 at the indicated concentrations in a buffer containing 35 mM Tris-Acetate (7.5), 50 mM KCl, 7 mM magnesium acetate, 150 mM AMPPNP, 50 µg/mL BSA, 1 mM TCEP-HCl, 10% glycerol, and 0.05% NP-40 for 1 hr at room temperature. The beads and supernatant were separated by centrifugation and the beads were washed twice with buffer to remove non-specific binding. The pulled-down protein complexes were eluted and separated through electrophoresis. The protein bands were visualized through immunoblots since GST-Rad54 (anti-GST; GE healthcare #27-4577-01, lot# 9610802) and Srs2 (anti-Srs2 (yC-18); Santa Cruz #sc11991, lot# C0905) were very similar in size.

## ATPase assay

A coupled spectrophotometric ATPase assay was performed in D-loop reaction buffer to keep consistency, which contains 35 mM Tris-acetate pH 7.5, 50 or 100 mM NaCl, 7 mM magnesium-acetate, 2 mM ATP, 1 mM DTT, 0.25 mg/mL BSA, and an ATP regenerating system (30 U/mL pyruvate kinase (Sigma), 3 mM phosphoenolpyruvate (Sigma), and 0.3 mg/mL NADH (Sigma)), as described before with slight modification (*Liu et al., 2006*). 10 µM (in nucleotides) φX174 ssDNA was included in the buffer or not as cofactor, and 5 nM Srs2 was added to initiate the reaction at 30°C. Only the linear portions of time courses from the absorbance decreases at 340 nm were used to calculate ATP hydrolysis rates.

## MMS and UV sensitivity assays

Serial dilution survival assay was carried out as published (*Liu et al., 2011*). Indicated strains (*Table 1*) were used to inoculate 5 mL YPD cultures and grown overnight at 30°C. Cultures were then diluted to $OD_{600\ nm} = 2.0$ with six sequential 5-fold dilutions. 2 μl of each culture dilution was plated on YPD with or without MMS (0.02%, 0.04%) or UV exposure (1 J/m$^2$) and incubated for 2 days at 30°C before imaging.

## Determination of damage induced recombination frequencies

The assay was carried out exactly as described before (*Ede et al., 2011*). Briefly, overnight cultures of different strains (*Table 1*) were diluted into 6 ml of $1 \times 10^7$ cells/mL and agitated at 30°C for 1.5 hr before the addition of α–factor to a final concentration of 10 μg/mL. After 1.5 hr, more α-factor was added into the culture to reach a final concentration of 13.3 μg/mL and incubated for one more hour before harvesting. Cells were resuspended and incubated with SC-ura media containing 0, 0.04, 0.08, 0.16, 0.24 μg/mL 4-Nitroquine-N-oxide (4-NQO) for 105 min at 30°C with agitation. By this time cells have traversed S-phase and were collected, washed, and plated onto YPD to determine viable titer and G418 plates to determine the G418$^R$ recombinants. Spontaneous rates were measured as before (*Ede et al., 2011*) using the method of the median (*Lea and Coulson, 1949*).

## Acknowledgements

We are grateful to Giordano Liberi, Wilfried Kramer, and Hanna Klein for sending strains, to Patrick Sung for sending the Srs2 expression plasmid, and to Peter Burgers for providing DNA Polymerase δ, PCNA, and RFC, which were supported by award GM32432 to PB. We thank Steve Kowalczykowski, Paula Cerqueira, Aurèle Piazza, Hang Phuong Le, João Vieira da Rocha, and Shanaya Shah for helpful comments on the manuscript. This work was supported by a postdoctoral fellowship ED 162/1–1 from the Deutsche Forschungsgemeinschaft (CE), and grants CA187561 (JL), GM081433 (MTW), CA92276 (WDH) and GM58015 (WDH) from the National Institutes of Health.

## Additional information

### Funding

| Funder | Grant reference number | Author |
| --- | --- | --- |
| National Institutes of Health | CA187561 | Jie Liu |
| Deutsche Forschungsgemeinschaft | ED 162/1-1 | Christopher Ede |
| National Institutes of Health | GM081433 | M Todd Washington |
| National Institutes of Health | GM58015 | Wolf-Dietrich Heyer |
| National Institutes of Health | CA92276 | Wolf-Dietrich Heyer |

The funders had no role in study design, data collection and interpretation, or the decision to submit the work for publication.

### Author contributions

JL, Formal analysis; Funding acquisition; Investigation; Writing—original draft; Writing—review and editing; CE, SSJ, Investigation; Writing—original draft; WDW, Formal analysis; Investigation; Writing—original draft; Writing—review and editing; SKG, Investigation, Writing—review and editing; BDF, Resources; Approved final submitted version Todd; MTW, Resources, Approved final submitted version; XV, Resources; Writing—original draft; Approved final submitted version; W-DH, Conceptualization, Formal analysis, Supervision, Funding acquisition, Writing—original draft, Project administration, Writing—review and editing

### Author ORCIDs

Wolf-Dietrich Heyer, http://orcid.org/0000-0002-7774-1953

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
