## [Decision Letter]

Thank you for submitting your article "Srs2 Promotes Synthesis-Dependent Strand Annealing by Disrupting DNA Polymerase δ-Extending D-loops" for consideration by *eLife*. Your article has been favorably evaluated by John Kuriyan (Senior Editor) and three reviewers, one of whom is a member of our Board of Reviewing Editors. The reviewers have opted to remain anonymous.

The reviewers have discussed the reviews with one another and the Reviewing Editor has drafted this decision to help you prepare a revised submission.

Summary:

This manuscript describes the biochemical reconstitution of a significant part of the SDSA homologous recombination pathway in which the Srs2 a 3'-5' helicase with pro and anti-recombination roles in vivo, participates. The authors explore the pro-recombination role of Srs2 by establishing robust assays for D-loop formation and dissociation in vitro with purified components. Revealing this essential pro-recombinogenic role of Srs2 in SDSA is of general importance for the genome stability field. Srs2 is able to displace D-loops both un-extended and extended by SUMO-PCNA and DNA polymerase δ. The authors show that Srs2 is highly active in dissociating D-loop generated by Rad51 and Rad54 with a 607 nucleotide invading ssDNA. They also show Srs2 has a weaker activity in disrupting extended D-loops made in reactions that include DNA polymerase δ/RFC/PCNA. The use of different forms of PCNA allowed the authors to show that it is the SUMO-PCNA extended D loops the one affected by Srs2. No effect is seen with Ubi-PCNA or wild-type PCNA suggesting that only the SUMO-PCNA form is able to promote extension that is displaced by Srs2. The manuscript is clear and extremely well written. It just needs some clarifications and some experimental modifications, as suggested below, to strengthen some conclusions.

Essential revisions:

The data presented in Figure 1, Figure 2 and Figure 4 provide convincing support of D-loop dissociation by Srs2, but some of the later analyses are less convincing. In Figure 3, Srs2 disruption of protein free and protein-bound D loops is compared. The authors state that in order to directly compare the activity of Srs2 on RecA-generated vs. Rad51-generated D-loops they needed to change the buffer conditions, but this seems to result in a very poor yield of D-loops in the Rad51 reaction (~12% compared with >50% in Figure 1). Without Srs2 the level of D-loops continues to increase over the time course. The concern here is that addition of Srs2 is preventing formation of D-loops in the ongoing reaction instead of by D-loop disruption. By contrast, the RecA reaction seems to be complete at the time Srs2 is added (yield of D-loops is not changed in the absence of Srs2); thus the drop in D-loop yield observed after Srs2 addition probably reflects D-loop disruption. It is not clear from the data presented that Srs2 is better at disrupting D-loops made by Rad51 over those made by RecA.

Analysis of D-loops extended by DNA polymerase δ/RFC/PCNA is complicated by the need to use a shorter oligo substrate to avoid topological constraints on extension of the invading strand. Thus, the yield of D-loops and extended D-loops with a 100 nt ssDNA is very similar to that reported previously by the Krejci lab. The Krejci and Sung groups have previously shown that Mph1 helicase is highly active in dissociating unextended as well as extended D-loops, but these two groups observed no or only weak D-loop disruption by Srs2. Here, the authors report a very modest effect of Srs2 in the reconstituted system, but again the analysis is complicated by the potential for competing reactions. The yield of extended D-loops increases over the time course in the absence of Srs2 raising the question of whether Srs2 is preventing D-loop extension (by its strippase function or by preventing access to Pol δ) or disrupting extended D-loops after they are made. This is particularly problematic for interpreting the data with SUMO-PCNA because the yield of unextended D-loops is higher in the reactions with SUMO-PCNA than with WT PCNA suggesting SUMO-PCNA is inhibiting D-loop extension rather than preferentially disrupting extended D-loops. It would be necessary to start with conditions where all D-loops have been extended before adding Srs2 to ensure only the disruption activity is monitored as well as to assess the role of the Srs2 ATPase activity in disruption of extended D-loops, especially with SUMO-PCNA.

---

## [Author Response]

*Summary:*

This manuscript describes the biochemical reconstitution of a significant part of the SDSA homologous recombination pathway in which the Srs2 a 3'-5' helicase with pro and anti-recombination roles in vivo, participates. The authors explore the pro-recombination role of Srs2 by establishing robust assays for D-loop formation and dissociation in vitro with purified components. Revealing this essential pro-recombinogenic role of Srs2 in SDSA is of general importance for the genome stability field. Srs2 is able to displace D-loops both un-extended and extended by SUMO-PCNA and DNA polymerase δ. The authors show that Srs2 is highly active in dissociating D-loop generated by Rad51 and Rad54 with a 607 nucleotide invading ssDNA. They also show Srs2 has a weaker activity in disrupting extended D-loops made in reactions that include DNA polymerase δ/RFC/PCNA. The use of different forms of PCNA allowed the authors to show that it is the SUMO-PCNA extended D loops the one affected by Srs2. No effect is seen with Ubi-PCNA or wild-type PCNA suggesting that only the SUMO-PCNA form is able to promote extension that is displaced by Srs2. The manuscript is clear and extremely well written. It just needs some clarifications and some experimental modifications, as suggested below, to strengthen some conclusions.

Thank you for the nice summary. Upon quantitation and calculating D-loop dissociation rates, it actually appears that Srs2 has an about two-fold preference to disrupt extending D-loops over D-loops made with the 607 nt substrate (compare Figure 1 with Figure 5). We added this information to the manuscript. In addition, we want to clarify that Srs2 can disrupt extending D-loops with all forms of PCNA (SUMO, ubiquitin, unmodified) but shows a slight preference for D-loops extending with SUMO-PCNA.

*Essential revisions:*

*The data presented in Figure 1, Figure 2 and Figure 4 provide convincing support of D-loop dissociation by Srs2, but some of the later analyses are less convincing. In Figure 3, Srs2 disruption of protein free and protein-bound D loops is compared. The authors state that in order to directly compare the activity of Srs2 on RecA-generated vs. Rad51-generated D-loops they needed to change the buffer conditions, but this seems to result in a very poor yield of D-loops in the Rad51 reaction (~12% compared with >50% in Figure 1). Without Srs2 the level of D-loops continues to increase over the time course. The concern here is that addition of Srs2 is preventing formation of D-loops in the ongoing reaction instead of by D-loop disruption. By contrast, the RecA reaction seems to be complete at the time Srs2 is added (yield of D-loops is not changed in the absence of Srs2); thus the drop in D-loop yield observed after Srs2 addition probably reflects D-loop disruption. It is not clear from the data presented that Srs2 is better at disrupting D-loops made by Rad51 over those made by RecA.*

We agree with this comment about the RecA experiments and decided to eliminate these experiments. The reaction conditions of the bacterial RecA reaction and the yeast Rad51/Rad54 reaction are quite different. For example, the RecA reaction is sensitive to salt, whereas the Rad51/Rad54 reaction is stimulated by salt. Moreover, the RecA reaction shows different kinetics and yield making a direct comparison difficult. We tried many conditions to compare the two enzymes, both in their respective optimal and permissive condition. In hindsight, we should have omitted these experiments from the manuscript, but we had invested so much that we initially decided to include them. We think that we can eliminate the RecA experiments and still maintain that Srs2 shows specificity for the reconstituted reaction with yeast proteins, as D-loop disruption in the reconstituted assay (see Figure 1) is ten-fold better than the disruption of protein-free D-loops (Figure 3) [see subsection “Srs2 disrupts Rad51/Rad54 produced D-loops more effectively than protein-free D-loops”].

*Analysis of D-loops extended by DNA polymerase δ/RFC/PCNA is complicated by the need to use a shorter oligo substrate to avoid topological constraints on extension of the invading strand. Thus, the yield of D-loops and extended D-loops with a 100 nt ssDNA is very similar to that reported previously by the Krejci lab. The Krejci and Sung groups have previously shown that Mph1 helicase is highly active in dissociating unextended as well as extended D-loops, but these two groups observed no or only weak D-loop disruption by Srs2. Here, the authors report a very modest effect of Srs2 in the reconstituted system, but again the analysis is complicated by the potential for competing reactions. The yield of extended D-loops increases over the time course in the absence of Srs2 raising the question of whether Srs2 is preventing D-loop extension (by its strippase function or by preventing access to Pol δ) or disrupting extended D-loops after they are made. This is particularly problematic for interpreting the data with SUMO-PCNA because the yield of unextended D-loops is higher in the reactions with SUMO-PCNA than with WT PCNA suggesting SUMO-PCNA is inhibiting D-loop extension rather than preferentially disrupting extended D-loops. It would be necessary to start with conditions where all D-loops have been extended before adding Srs2 to ensure only the disruption activity is monitored as well as to assess the role of the Srs2 ATPase activity in disruption of extended D-loops, especially with SUMO-PCNA.*

Thank you, this has been a really helpful comment and pushed us to strengthen the conclusion that Srs2 can dissociate extending D-loops with several new experiments, involving also a new experimental design. First, as the reviewers correctly note, in the old Figure 5 data D-loop extension was not completed and extended D-loops increased during the Srs2 time course, but less so in the presence of Srs2 than in its absence. In the new Figure 5, parts A-C, we kept the original experimental design with an end-labeled oligonucleotide as invading DNA but increased the incubation time with the polymerase from 6 to 15 min to drive most D-loops into the extended form. Addition of increasing amounts of Srs2 clearly shows a reduction of extended D-loops from about 30% to about 10%. This is accompanied by a congruent increase in the expected products of collapsed extended D-loops visualizing the extended, end-labeled invading DNA. Second, we added in the new Figure 5, parts D and E, a new experimental design producing radiolabeled extended D-loops using *α*^[32]^P-dCTP incorporation. The time course after addition of 25 nM Srs2 shows a dramatic decrease in extended D-loops and a concomitant increase in the expected signal for the radiolabeled, extended invading strand of the collapsed extended D-loop. This experiment has a more qualitative character and the quantitation is provided as a supplement (Figure 5—figure supplement 1).

In the experiments of Figure 5, we did not observe a significant difference between SUMO- PCNA and unmodified PCNA. This is likely related to the fact that the reaction conditions were not designed to document subtle differences. For example, Srs2 was not limiting and the incubation temperature was optimal. In the new Figure 6, we use the *α*^32^P-dCTP incorporation assay and a titration of Srs2 at a lower temperature (25 instead of 30 °C) to slow the reaction. Here, we see a slight, but reproducible and significant, preference for disrupting D-loop extending with SUMO-PCNA. In these reactions, higher amounts of Srs2 (25 nM) still disrupt all D-loops efficiently (Figure 6—figure supplement 1). This slight preference for SUMO-PCNA extending D-loops seen in Figure 6 is congruent with the results from Figure 7 using the end-labeled invading DNA substrate in the D-loop reaction (original Figure 6), and these effects depend on the presence of Srs2 (Figure 7—figure supplement 1) and active extension (i.e. presence of dNTPs; Figure 7—figure supplement 2). We are careful in our wording to reflect that Srs2 has only a slight preference under our assay conditions for SUMO-PCNA over unmodified and UBI- PCNA.